

**Quality assessment of the TOPAZ4 reanalysis in**
**the Arctic over the period 1991-2013**
**Jiping Xie**[1]**, Laurent Bertino**[1]**, Francois Counillon**[1]**, Knut A. Lisæter**[1]**, and Pavel**
**Sakov**[2]
[1]Nansen Environmental and Remote Sensing Center, Bergen N5006, Norway
[2]Bureau of Meteorology, Melbourne VIC3001, Australia
**E-mail:** jiping.xie@nersc.no
**Abstract** Long dynamical atmospheric reanalyses are widely used for
climate studies, but data assimilative reanalyses of the Arctic ocean and sea
ice are less common. TOPAZ4 is a coupled ocean and sea ice data
assimilation system for the North Atlantic and the Arctic that is based on the
HYCOM ocean model and the Ensemble Kalman Filter data assimilation
method using 100 dynamical members. A 23-years reanalysis has been
completed for the period 1991-2013. This study presents its quantitative
quality assessment, compared to both assimilated and unassimilated
observations available in the whole Arctic region in order to document the
strengths and weaknesses of the system for potential users. It is found that
TOPAZ4 performs well with respect to near surface ocean variables, but



some limitations appear in the interior of the ocean and for ice thickness,
where observations are sparse. In the course of the reanalysis, the skills of
the system are improving as the observation network becomes denser, in
particular during the International Polar Year. The online bias estimation
successfully maintains a low bias in our system.
**Keywords**: Arctic Ocean, Reanalysis, HYCOM, EnKF.

## 8 1. Introduction

The Arctic Ocean plays an important role in the global climate system, and
in particular the sea ice at the interface between atmosphere and ocean
regulates the fluxes of heat, moisture and momentum in ice-covered regions.
The recent warming of the Arctic and the change of its water cycle has been
linked to the following manifestations: a significant reduction and thinning of
the sea ice cover (Johannessen et al., 2004; Shimada et al., 2006; Rothrock
et al., 2008; Kwok and Rothrock, 2009); more freshwater in the Arctic in the
2000s (Haine et al., 2015); more mobility and faster deformations of the Arctic
sea ice (Rampal et al., 2009; Spreen et al., 2011). The interpretation of such
changes is severely hampered by the sparseness of the observations of sea
ice and ocean, which is not expected to improve dramatically in a near future.
It can be assisted by free-running model simulations, but those are usually
hampered by the mislocation of the ice edge, of certain water masses in an
unconstrained simulation. One then recurs to studying surrogate locations
where similar processes are assumed to take place.
The latter activities thus necessitate state-of-the-art reanalysis databases
able to honour accurately the observations in a physically consistent manner.
Recent efforts in Arctic Ocean state estimation have delivered either long-



window optimizations (Nguyen et al., 2009, 2011) or more often short-window
estimations (Schweiger et al., 2011; Mathiot et al., 2012; Sakov et al., 2012,
Chevallier et al. 2016). Long-window optimizations deliver continuous model
trajectories, which are physically more consistent than those using short
windows. On the other hand, slicing the optimization problem into short
windows makes the estimation problem more linear or better-conditioned
(fewer unknowns and observations) and delivers more accurate products.
Besides the window length, the choice of a background covariance matrix is
also a critical aspect in a data-scarce area such as the Arctic. The
background error covariance used in an ocean data assimilation system can
be – by increasing order of complexity - based on fixed multivariate spatial
statistics (Cummings et al., 2009), or an empirical estimation by a time-
invariant ensemble (Oke et al., 2008) or a seasonally variable ensemble
(Brasseur et al., 2005; Xie et al., 2011). In the case of ice-ocean systems, sea
ice data assimilation often relies on rudimentary ice-only nudging methods
(Schweiger et al., 2011; Tietsche et al., 2013), however the possibility to
account for flow-dependent coupled ice-ocean data assimilation updates had
already been demonstrated in Lisæter et al. (2003). The Pilot TOPAZ4
reanalysis of Sakov et al. (2012) has shown that the forecast error covariance
from a dynamical ensemble was mitigating the physical inconsistencies that
could be expected from a short assimilation window.

22       The TOPAZ4 system is a coupled ocean-sea ice data assimilation system

of the physical environment in the North Atlantic and Arctic Ocean, which was
initially used for short-term forecasting (Bertino and Lisæter, 2008) and later
on for reanalysis (Sakov et al., 2012). TOPAZ4 represents the Arctic
component of the MyOcean system (marine.copernicus.eu) where it is also



used with coupling to an ecosystem model (Samuelsen et al., 2015; Simon et
al., 2015). The present paper follows the Pilot TOPAZ4 reanalysis by Sakov et
al. (2012), which demonstrated the performance of the same system for the
period of 2003-2008. They found that the EnKF data assimilation method can
avoid ensemble collapse, and that the ensemble statistics provide reliable
state-dependent error estimates and that data assimilation improves the
match to independent observations compared to a free model.
TOPAZ4 system is forced by the ECMWF ERA Interim reanalysis (Dee et
al., 2011) and assimilates most available measurements including along-track
altimetry data, sea surface temperatures, sea ice concentrations and sea ice
drift data from satellites as well as in situ temperature and salinity profiles.
The proposed reanalysis is four times longer (1991-2013) than the pilot
reanalysis and includes periods with poor observations and more intense
observations efforts, such as during the International Polar Year (IPY, 2007-
2009). The focus of this study is to provide a quantitative assessment of the
performance of the reanalysis for ocean and sea ice variables in the pan-
Arctic region (defined as north of 63°N) in order to guide the user about its
skills and limitations. In particular, we investigate the variability of the
performance in space, its seasonal cycle and its trend in the course of the
reanalysis.
The outline of this paper is as follows: In section 2, the reanalysis system is
described including the model, the data assimilation scheme, and the
implementing set-up. Section 3 presents the result of against available
observations: altimetry, SST, T-S profiles, ice concentration ice drift and ice
thickness. For each of these quantities we assess variability of the
performance of the system with space or with time. In Section 4, we



summarize the results, and discuss the results and the potential ways to
improve our system for the next version of the reanalysis.
**2. The reanalysis system**
**2.1 The HYCOM ice-ocean model**
The TOPAZ4 system uses version 2.2 of the Hybrid Coordinate Ocean
Model (HYCOM) developed at University of Miami (Bleck, 2002; Chassignet
et al., 2003). The model is mainly unchanged compared to that used in Sakov
et al. (2012). It uses 28 hybrid z-isopycnal layers, and the top layer has a
minimum thickness of 3 m. The model grid has a horizontal resolution of 12-
16 km, which is eddy permitting from the Equator to the Nordic Seas but is still
far from being eddy-resolving in the Arctic. The lateral boundaries of
temperature and salinity are relaxed to a combination of the World Atlas of
2005 (WOA05, Locarnini et al., 2006) and the version 3.0 of the Polar Science
Center Hydrographic Climatology (PHC, Steele et al., 2001). HYCOM is
coupled to a sea ice model in which the ice thermodynamics are described in
Drange and Simonsen (1996) and the elastic-viscous-plastic rheology in
Hunke and Dukowicz (1997). The surface momentum fluxes use a bulk
formula parameterization (Kara et al. 2000), and the related thermodynamic
fluxes are computed as described in Drange and Simonsen (1996).
The model has been initialized from the same climatology data as used at
the boundaries. The model is spin up from 1973 using the European Center
for Medium-Range Weather Forecast (ECMWF) ERA-Interim reanalysis data
(Simmons et al., 2007). The Pacific water inflow is imposed by a barotropic
inflow through the Bering Strait at the model boundary and balanced by an out
flow at the southern boundary of the domain. Unlike in Sakov et al. (2012), the



inflow varies seasonally as found in observations (Woodgate et al., 2005):
with a maximum in June (1.3 Sv), a minimum in January (0.4 Sv), and the
mean transport of 0.8 Sv.
**2.2 Data assimilation with the EnKF**
Given observations, a model forecast, and assumptions on their
respective uncertainties and at time $t_i$, the analyzed model state and its
uncertainty can be computed by data assimilation using the least squares
minimization (Evensen, 1994, 2003):

10$$\mathbf{X}_i^a = \mathbf{X}_i^f + \mathrm{K}_i(\mathbf{d_i} - \mathbf{HX}_i^f) \tag{1}$$

Where $\mathbf{d_i}$ is the observation vector, $\mathbf{X}_i$ is the model state vector and $\mathbf{H}$ is the
observation operator denoting the projection from the model state variables to
the measurement. The superscripts "a" and "f" refer respectively to the
analyzed and the forecast states. We use the Deterministic form of the EnKF
(Sakov and Oke 2008). Compared to Sakov et al. (2012), the only
modification is the removal of the 1% multiplicative inflation near the end of
the reanalysis (January 2010), which becomes problematic when used with
spatially varying observational network (Anderson et al, 2001). Hence,
multiplicative inflation in absence of observation (such as in the interior of the
Arctic Ocean) leads to an exponential increase of the spread, which combined
with a multivariate update makes it very sensitive to bias in the observed
variable. Satellite sea ice concentration interpret melt pond in model as open
water, which are not considered in TOPAZ4. This bias in the observed
variable, was leading to a degradation of the stratification in the Arctic. The
bias estimation procedure has also been modified as explained below (see
Section 2.4).



## 2.3 Assimilated observations

The observations assimilated into the TOPAZ4 system are the same types as those assimilated in Sakov et al. (2012) except for some updates in the data sources. They are the satellite Sea Surface Temperature (SST), along-track Sea Level Anomalies (SLA) from satellite altimeters, in situ temperature and salinity profiles, Sea Ice Concentration (ICEC) and low-resolution sea ice drift data from satellites. An overview of the observations used in the reanalysis is given in Table 1. The preprocessing, temporal averaging and observation errors are following the procedure described in Sakov et al. (2012).

The SST data are initially the 1° resolution Reynolds SST from NOAA (Reynolds and Smith, 1994), upgraded in June 1998 by the high-resolution OSTIA data (Stark et al, 2007) from UK Metoffice. The SLA data assimilated in this system are the delayed time product (vxxc), which is validated unfiltered and not sub-sampled along-track SLA data with a resolution of 7 km, by Collecte Localisation Satellites (CLS). Furthermore, the assimilated ICEC are from the Ocean & Sea Ice Satellite Application Facility (OSISAF). Before 19 June 2002, this product is derived from SSM/I at 25 km resolution, and later derived from ARMSR-E 89 GHz brightness temperature at 12.5 km resolution. In the last three years, the ICEC resolution has been upgraded to 10 km. The temperature and salinity profiles include Argo floats, Damocles Ice-Tethered Profiles (ITP) and a large collection of hydrographic cruise data, as described in Sakov et al. (2012). At the exception from Reynolds SST, all data sources are available through the MyOcean portal.



**2.4 Bias estimation in the TOPAZ4 reanalysis**

As described in Sakov et al. (2012), two bias fields (for SST and mean sea surface height (MSSH)) are estimated online by model state augmentation, so that the analysis state of Equation (1) is modified as:

$$\begin{pmatrix} \mathbf{X}_i^a \\ \mathbf{B}_i^a \end{pmatrix} = \begin{pmatrix} \mathbf{X}_i^f \\ \mathbf{B}_i^f \end{pmatrix} + \mathrm{K}_i(\mathbf{d_i} - \mathbf{HX}_i^f + \mathbf{HB}_i^f) \, , \qquad (2)$$

where $\mathbf{B_i^a}$ represents the bias estimates at the analysis time. The initial biases for each ensemble member are homogeneous in space, uniformly distributed with random values. For the SST, the initial biases are taken in [-4, 4] °C. And for the MSSH, the initial biases are in [-0.6, 0.6] m.

The bias fields are updated according to the sample covariance from the forecast ensemble and do not evolve during model integration. To avoid a collapse of the bias ensembles, an adequate multiplicative inflation is used (2% for SLA and 6% for SST). The multiplicative inflation of bias did not handle well the changes of observations coverage and has been replaced in May 2006 by additive inflation of identical amplitude (here after calling the event E9), using an auto-regressive temporal process of order one. After several assimilation steps, the bias fields converge to temporally stable but spatially variable fields. Figure 2 shows the bias estimates at end of the reanalysis for the SSH and the SST in the Arctic region. The bias patterns compare well with those obtained in Sakov et al. (2012)[1]. There are small discrepancies because the bias is estimated at a different time - December 2009 in Sakov et al. (2012) instead of December 2013 here - and the bias estimation is the result of a longer estimation period for which the signal to

---

[1] Sakov et al. (2012) present the mean SSH bias while we show the SSH bias of opposite sign.



noise ratio is reduced. The RMSD using the online-bias corrected values are
slightly lower than the bias estimate of the last analysis step (not shown).
Although the static part of the bias would be better estimated thoretically at
the last assimilation of the reanalysis, the online bias approach can handle
errors in the trend from seasonal bias and adjusting the observational network.
The online bias estimate is provided together with the model output from 1999.
In the following validation sections, the online bias estimates $\mathbf{B_i^a}$ are used to
offset the reanalysis state from that date.
In order to avoid inconsistencies between assimilation of SST and
temperature profile, the SST bias is injected downwards into the model mixed
layer and decays exponentially.
## 3. Quantitative accuracy
In this section, we investigate whether the accuracy of the reanalysis
varies spatially, seasonally or interannually. Such information is necessary for
potential users of the reanalysis product. It also pinpoints the model limitations
that motivate further model developments. The model error is calculated from
the daily averages of the ensemble mean and the observations. The misfits of
bias and root mean square differences (RMSD) are calculated as described in
Equations of (3) and (4):
$$\text{Bias} = \frac{1}{N}\sum_{i=1}^{N}(\mathbf{H_i\overline{X}_i^f} - \mathbf{d_i} - \mathbf{HB_i^f}) \qquad\qquad (3)$$
$$\text{RMSD} = \sqrt{\frac{1}{N}\sum_{i=1}^{N}(\mathbf{H_i\overline{X}_i^f} - \mathbf{d_i} - \mathbf{HB_i^f})^2} \quad , \qquad (4).$$
Where $\mathbf{\overline{X}_i^f}$ is the forecasted daily average from the ensemble mean, which is
compared to the observations $\mathbf{d_i}$ on the same day. N is the number of daily



averages available over the diagnostic period (like either 365 or 366 for
yearly). For SST and SLA, the bias term of $B_i^f$ is the online bias estimate
($B_i^f = B_{i-1}^a$). Error bars are used to represent the standard deviations of these
quantities - i.e. the variability of the RMSD or bias estimate through the
calculation period.
**3.1 Sea Level Anomalies**
The variability in the SLA reanalysis accuracy is evaluated in the Pan-
Arctic region (defined to the North of 63°N, see Fig. 1). The spatial variability
of the bias and RMSD, calculated over the whole reanalysis period (1993-
2013), is shown to the top of Fig.3. The residual bias is mainly positive, with
much smaller amplitude than the estimated bias (see Fig.2). Some positive
bias features reach up to 6 cm along the sea ice edge in the Greenland Sea
and south of the Baffin Bay. Except for the sea ice edge in the Greenland Sea,
the high RMSDs of SLA over 10 cm match the areas of large bias shown in
Fig. 2. The spatially averaged bias is 2.1 cm, and the RMSD is about 6.7 cm.
The yearly time series of the SLA misfits and the observation number are
shown in the left column of Fig. 4. The number of assimilated observations
evolves with the launch or completion of satellite missions. The number of
observation increases in 2000 with the launch of the GEOSAT Follow On
(GFO) mission. The missions of Topex, Jason 1 and Jason 2 do not
contribute directly in the Pan-Arctic region as their inclination is 66°, unlike 70°
for GFO. There is a low observation period in 2008-2009 with the end of GFO
mission, follow by an increase in 2011 with Cryosat-2, a decrease in 2012
with the end of Envisat and an last increase with the Saral/AltiKa mission in



2013. From 1993 to 2013, the RMSD decreased gradually from 8 cm to less
than 6 cm. The most notable change is the introduction of the bias estimation
and its reset in 2006. Afterwards, the residual bias stabilizes around 1cm but
remains positive. The RMSD reduces then gradually with the introduction of
new and more accurate observations. The reduced altimeter constellation in
2008-2009 does not cause an increase of the errors. This demonstrates the
advantage of assimilating multiple types of observations, as improved SSH
may also be the result of improved SST, or temperature and salinity profiles.
Meanwhile, the temporal standard deviation of the RMSD during the year
(shown as the-half-error bar) also reduces from 1-2 cm to less than 1 cm,
indicating the system is getting more stable with time.
The seasonal cycle of the accuracy is shown in the right column of Fig. 4. The
SLA being masked by sea ice, the number of observations varies seasonally
in opposition to the sea ice cover. There is some seasonality in the RMSD,
ranging from 5.5 to 7 cm as a consequence of the seasonal spatial coverage.
The residual bias is positive through of the whole year reaching a maximum in
April. This may be explained as well by the seasonal sea ice coverage, but
also by a possible underestimation of the thermal expansion. The standard
deviations of the residual bias and RMSD have no visible seasonality.
**3.2 Sea Surface Temperatures**
The spatial distribution of the SST misfits during 1999-2013 is shown in
the bottom panels of Fig. 3. Note that SST is also masked under sea ice, as
done during assimilation. There are stripes of cold residual bias and high
RMSD along the ice edge from the North of the Barents Sea until the South of
the Greenland Sea. These are contradictory to the sea ice concentration



biases in the same areas in Section 3.4, where a cold bias corresponds with
too little ice. The accuracy of SST observations near the ice edge is poor and
relies on strong ad-hoc assumptions. Another salient feature is the warm bias
(> 0.2 °C) North of Denmark Strait. It is known that the recirculation of Atlantic
Water inflow there in TOPAZ4 is excessive as identified in *Lien et al. (2016)*.
This pattern was also visible in the estimated bias (Fig 2., with an opposite
sign), suggesting that the estimated bias account for most of the bias but that
it still underestimates the true bias. An additional stripe of the cold residual
bias and higher RMSD is clear along the Mohn's Ridge, also pointing to
topographic steering issues. In the Barents Sea, a weak bias is noticeable.
Besides these areas, most of the SST RMSD is lower than 0.5 °C. On
averaged in the whole Arctic region, the SST RMSD is about 0.45 °C during
the period 1999-2013.
The evolution of SST accuracy of the TOPAZ4 reanalysis is shown on the
left column of Fig. 5, together with the number of observations. In June 1998,
the coarse resolution Reynolds SST is swapped to the higher resolution
OSTIA SST and the number of observations increases drastically. On
average over the whole period 1991-2013, the SST RMSD is about 0.64 °C,
and the bias -0.08 °C. In the first years, the SST RMSDs are initially about 1
°C but decrease gradually down to 0.8 °C before the switch to OSTIA in 1998.
During that period, the model has a cold SST bias around -0.3 °C and a
standard deviation of 0.1 °C. After the introduction of OSTIA SST in 1998, the
SST bias settles down closer to zero, but a slight positive is still noticeable
before 2011. The RMSD decreases rapidly below 0.6°C in 1998 as a direct
consequence of the reduction of the bias and the more abundant
observations. The RMSD continues to reduce and reaches a minimum in



2010 below 0.4°C. At that time, the ensemble spread was getting too small,
and the system performance was too constrained by SST as can be seen on
the standard deviation of RMSD. It was thus decided to increase artificially the
SST observation errors, which resulted in a small increase of the error up to
0.5 °C. It is clear from the above that the transition to high-resolution SST in
our system has led to higher SST accuracy.

7       Furthermore, the seasonal variability of the SST performance is shown in

Fig. 5. As for SLA, the number of observations varies seasonally with the sea
ice mask and causes changes of the bias and RMSD: the RMSD is minimum
in September and October with less than 0.4 °C when there are more
observation and is maximum over 0.6 °C in June and July when the bias is as
well maximum. The reason for the larger bias in summer months is unclear
but is possibly related to errors in the mixed layer depths and the atmospheric
radiative forcing.

### 3.3 In situ temperature and salinity profiles

17       There are $1.1 \times 10^5$ temperature and salinity profiles assimilated in the Pan-

Arctic region during the period 1991-2013, but the distribution of these profiles
is very uneven both in time and space, with more observations in ice-free
areas and during the IPY. The averaged assimilated temperature and salinity
profiles, the corresponding profiles from the reanalysis, the climatology
WOA13 (Locarnini et al., 2013) and their misfits are all shown in Fig.6. The
stratification in the Arctic varies regionally, depending on the influence of the
Atlantic Water inflow, the Pacific Water inflow and the river discharge. We
have thus divided the analysis into four sub-regions: the central Arctic, the
Barents Sea, the Greenland Sea, and the Norwegian Sea (see Fig.1).


In the central Arctic, the average profiles depict well the cold halocline
water near the surface and warmer water and saline water around 400 m
associated with Atlantic Water. In the near surface, TOPAZ4 is doing slightly
better than the climatology for salinity. The core Atlantic Water is clearly too
diffuse in TOPAZ4 (not pronounced enough and vertically too broad) leading
to a cold bias (-0.3 °C) and RMSD (0.5 °C) around that depth. An area of
larger RMSD is noticeable around 1000 m (0.6 °C and 0.3 psu). Since the
bias at that depth is low and since climatology has lower RMSD, it suggests
that TOPAZ4 has too much variability at depths. That variability is likely due to
the data assimilation setup with the combined effect of multiplicative inflation
and spurious correlations (see Section 2.2).
In the Greenland Sea, the temperature RMSDs and biases are again slightly
better than the climatology near the surface (upper 200m), but degrade very
near below, reaching the maxima of RMSD (> 1 °C and 0.1 psu) and bias
around 800m.
In the Norwegian Sea, the features are similar: the model having some skills
near the surface but deteriorating at depths.   The Atlantic Water (AW) is
present but it is too diffuse. It is too broad and does not capture the maximum
at the same depth as in the observation. It is a well-known limitation of ocean
models nowadays (Ilicak et al., 2016).
In the Barents Sea, the RMSD for temperature and salinity is improved
near surface, compared to the climatology. But the AW (temperature > 3°C
and salinity > 35 psu, Blindheim and Østerhus, 2003) of the TOPAZ4 is too
warm and saline, which suggests there is too much AW inflow or too weak
vertical mixing.



In Fig. 7, the evolution of the performance of TOPAZ4 for temperature and
salinity is presented by the time series of innovation diagnostics at depths of
300-800 m. The system performance is relatively stable until 2006, with a cold
and fresh bias and RMSD of about 1.5°C and 0.1 psu. From 2006, the
performance is greatly improved, supposedly by the combined effect of the
change from multiplicative to additive inflation and with the large increase of
observation number during the international Polar Year  (IPY) period. Once
the IPY period is finished, the number of observation drastically reduces. It
results in a clear increase of the RMSD and bias for temperature to a level
intermediate to prior the IPY period, but the error for salinity remains to a
lower level than before the IPY.
**3.4 Sea ice concentration**
Relative to the daily sea-ice concentration product from OSISAF
(Copernicus OSI TAC product), the spatial variability of the daily misfits of
sea-ice concentration is shown in Fig.8. As there is a large seasonal variability
in the sea-ice extent, this is carried out at two characteristic times of one year:
the maximum (March) and minimum ice extent (September).
In March, there is a dipole anomaly on either sides of the ice edge in the
Greenland Sea. The ice edge in TOPAZ4 is transiting too sharply from pack
ice to open water, because the heat capacity of the ice is neglected. This
leads to a dipole bias (positive inside the ice and negative outside) during the
melting season. There is also a weak bias over regions that are usually ice-
free. Indeed, OSISAF does not employ weather filtering and places a thick
band of low concentration (< 10%) in ice-free region (Ivanova et al. 2015).





In September, TOPAZ4 shows a negative bias in the Greenland Sea. At that
time of the year, the sea ice flows southwards and TOPAZ4 tends to
underestimate the southern extension of the sea ice tongue along Greenland.
This indicates that the dynamical forcing is biased or that the drag coefficients
are incorrect as the ice is in free drift there.
The RMSD is approximately 5% in the whole of Arctic region, which is good in
view the accuracy of the data set (~10%). There are regions where the RMSD
exceeds 25%, which coincides with regions where the bias is high. Data
assimilation does constrain the sea ice concentrations but the model biases
(lack of resolution of ocean currents, biases of ice drift or ice thickness) still
cause locally high residual errors of ice concentrations.
In order to assess the interannual variability of the performance of
TOPAZ4, we have decided to use the standard sea-ice extent (SIE) metric.
SIE is calculated as the surface area in which the ice concentration is larger
than 15 %.
As the variability in the decadal trend of SIE in the Arctic is large, we present
the interannual evolution in the whole Arctic and in two sub-regions: the
Greenland Sea and Barents Sea (Fig. 9). TOPAZ4 shows good agreement
with the OSISAF observations in the Pan-Arctic region and the mean SIE in
the 23 years are $8.03 \times 10^6$ instead of $7.96 \times 10^6$ km$^2$ in the observations. The
decreasing trend of the SIE during the period 1991-2013, is $-6.16 \times 10^4$ km$^2$ y$^{-1}$,
which compares well to the trend of the observations ($-6.34 \times 10^4$ km$^2$ y$^{-1}$).
In the Greenland Sea, the SIE in TOPAZ4 is underestimated, which clearly
relate the bias in the southern extent of the sea-ice tongue along the coast of
Greenland. The bias in TOPAZ4 is in averaged $-3.6 \times 10^4$ km$^2$ and the
decreasing trend in TOPAZ4 is $-3.1 \times 10^3$ km$^2$ y$^{-1}$, which is larger than observed



($-2.3\times10^3$ km$^2$ y$^{-1}$). In the Barents Sea, the variability agrees well, although
TOPAZ4 underestimates slightly the SIE. The decreasing trend is comparable.
The seasonality of the SIE in OSISAF and TOPAZ4 are investigated in Fig.
10. It is clear that the seasonal cycle of the ice extent is generally well
simulated by the reanalysis in the Pan-Arctic area. In the summer months
from June to August, a little underestimation of the extent in the reanalysis is
apparent, and the minimal ice extent is a little early compared to the
observations. In the Greenland Sea, the underestimation feature of sea ice
extent is large. The underestimation of sea ice extent starts in February and
increases during the sea ice melt, reaching a maximum (of about $1\times10^5$ km$^2$)
in July. In the Barents Sea, the seasonal cycle is well simulated but some
differences are noticeable there in the beginning of the year, reaching a
maximum in April, but close to zero in August and September when there is
no ice.
**3.5 Sea Ice Drift**
The sea ice drift from the buoy data of the International Arctic Buoy Probram
(IABP) are available at 12h frequency from 1991 to 2011,is an independent
data set and is used here for validation. To avoid the "survival bias" caused by
the retreat of sea ice from the marginal seas and unresolved coastal effects,
the buoy drift vectors are limited to the central Arctic, as shown with the red
line in the right panel of Fig. 1. The waters shallower than 30 m and closer
than 50 km from the coastline are excluded. This data set has been gridded to
be compared with the model. Each grid cell is filled (i.e. considered reliable) if
the calculation involves at least 30 buoys within a day. A coarser grid than the
model resolution is used (4 grid cells which corresponds to approximately



60x60 km$^2$) to avoid having too many empty cells. The daily averaged from
the measurement is the mean of the 12h drifting speed. For comparison, the
model drifting speed is calculated from daily averaged of eastward and
northward velocity. Several approximations are made during this comparison;
we compare Eulerian with lagrangian drift for which the drift is expected to be
larger; the model ice drift is calculated from daily average of u and v instead of
daily ice drift for which the drift is higher by approximately 0.5 km per day (not
shown).
On averaged over the period 1991-2011, the mean drift fields of sea ice are
presented in Fig. 11. As the resulting drift estimate appeared noisy, we have
applied a smoothing with the neighboring grid cells. Both observations and
TOPAZ4 show similar pattern with a pronounced Beaufort Gyre, although the
center of the Gyre differs slightly. We can also notice that TOPAZ4
overestimates globally the ice drift with a bias of 1.7 km d$^{-1}$. In the Chukchi
Sea, TOPAZ4 underestimates the drift by approximately -2 km d$^{-1}$.

16       Over the period 1991-2011, the monthly time series of the Ice drift

speeds are compared in Fig. 12. They are averaged in the Central Arctic from
the reanalysis and the buoy data respectively. On average, the drift speed is
about 7 km d$^{-1}$ in buoy data, and about 9.4 km d$^{-1}$ in TOPAZ4 reanalysis. The
fast bias is clear until the end of 2010. From that time, the drag coefficient of
the atmosphere on sea-ice has been reduced from 2.14x10$^{-3}$ to 1.6x10$^{-3}$. We
can see that the bias is much reduced during the last year. The RMSD is on
average 5.1 km d$^{-1}$, of which 2.5 km d$^{-1}$ can be attributed to the bias. The
correlation between the 2 curves is about 0.6. In addition, the monthly
seasonality anomaly of the ice drift for the period 1991-2011 is plotted in Fig.
13. While the buoys show a clear seasonality in the ice drift, being slowest in





March and fastest in September, the seasonality in TOPAZ4 reanalysis is weaker and reaches a minimum in May (delayed by 2 months).

### 3.6 Sea ice thickness

The sea ice thickness in Arctic has been paid much attention in recent years because it has been found to be sensitive to global warming (Kwok et al., 2009; Zygmuntowska et al., 2014). In this study, sea ice thickness is an independent data set, as it has not been assimilated. The observations of ice thickness are yet very few. A satellite-derived product for the Arctic Ocean ice provides the estimations of sea ice thickness for February-March and October-November between 2003-2008 (ICESat, Kwok et al., 2009). Figure 14 shows the spatial distributions of the mean sea ice thicknesses and their differences. The spatial correlations are 0.74 and 0.87 for spring and fall, respectively. On average, TOPAZ4 is too thin compared to ICESat with a bias of -0.79 m and -0.64 m, in spring and in fall. In spring, TOPAZ4 is too thin, in particular north of Ellesmere Island by approximately 2 m. There is a positive bias centered in the Beaufort Gyre in spring. In fall this bias is wider and displaced slightly to the east.

Another source of validation is the Unified Sea Ice Thickness Climate Data Record (Lindsay, 2013) resulting from a concerted effort to collect as many observations as possible of Arctic sea-ice draft, freeboard, and thickness. The sea ice draft is measured by Sonar of US Navy Submarines from National Snow and Ice Data Center (USSUB-DG and USSUB-AN, Wadhams and Horne, 1980; Wensnahan and Rothrock, 2005; Rothrock and Wensnahan, 2007), and the sea ice thickness by flight campaigns from NASA Operation IceBridge (IceBridge, Kurtz et al., 2013), as shown in Fig. 15(a). The sea-ice



draft data has been diagnosed in TOPAZ4 as proposed by the equation (4) of
Alexandrov et al. (2010):

$$D_i = H_i \cdot \frac{\rho_i}{\rho_w} + H_{sn} \cdot \frac{\rho_{sn}}{\rho_w} \qquad (5).$$

Where $D_i$ is ice draft, $H_i$ is ice thickness, and $H_{sn}$ is the snow thickness. The $\rho_i$,
$\rho_w$, and $\rho_{sn}$ are the densities for ice, water, and snow (respectively 900 kg m$^{-3}$,
1000 kg m$^{-3}$, and 300 kg m$^{-3}$).
The IceBridge ice thickness covers the period of 2009-2011. TOPAZ4
reanalysis is too thin with a bias of 1.1 m, a RMSD of 1.4 m and a correlation
of 0.5. The bias against the sea ice draft is smaller with 0.3-0.4 m, and a
RMSD about 0.6-0.7 m. The correlation coefficients are relatively good with
.86 and 0.69, which is higher than for the IceBridge data. These discrepancies
are likely to be related to the spatial distribution of the different data set.
Hence, IceBridge data is concentrated around the Northern coast of
Greenland where TOPAZ4 showed largest bias in the comparison with
ICESAT.
As another diagnostics of interest, the daily time series of sea ice volume from
TOPAZ4 in the Arctic of 1991-2013 is shown by the blue curve in the left
panel of Fig. 16. Before 2001, the sea ice volume varies stably around
1.4x10$^4$ km$^3$, with a significant seasonal variability between 8x10$^3$ km$^3$ and
1.9x10$^4$ km$^3$. Afterwards in the period 2001-2010, the sea ice volume
decreases dramatically. This reduction of sea ice volume is qualitatively
consistent with the limited satellite records. First the estimate from Kwok et al.
(2009), derived from the ICESat record from 2003 to 2008, shows a similar
trend. After revising the uncertainties of input data (snow depth, sea ice
density and ice concentrations), Zygmuntowska et al. (2013) corrected the





estimates of the mean sea ice volume, shown as the starred line in Fig. 16.
With respect to these sea ice volume estimates, TOPAZ4 still has too little ice.
In the right panel of Fig. 16, the seasonal cycles of sea ice volume from
TOPAZ4 and the standard deviation in the 23 years are shown by the blue
curve and the cyan error bars respectively. In May, the maximum sea ice
volume is about $1.5 \times 10^4$ km$^3$, and in September is less than $5 \times 10^3$ km$^3$. The
sea ice volumes from Zygmuntowska et al. (2013) are plotted on top of the
averaged TOPAZ4 seasonal cycle in the period 1991-2013. These
correspond well to the model climatology, but still betray an underestimation
because the measurements are representative of a period of lower ice volume.
The TOPAZ4 seasonal cycle of ice volume seems to change in amplitude
during different time eras, although the reasons lie in two successive changes
of the settings of the EnKF. In December 2001, the variance of precipitation
errors is increased from $1.10^{-17}$ to $1.10^{-12}$ m$^2$.s$^{-2}$, as an adjustment for a slow
decrease of ensemble spread. These perturbations being truncated Gaussian,
the truncation resulted in excessive snow precipitations. The excessive snow
depths has then isolated the ice from the atmosphere and reduced the
amplitude of the yearly cycle from 1.08 m to 0.74 m (see Figure 17), this also
delayed the phase of the cycle. In January 2011, an unbiased log-normal law
replaces the truncated Gaussian perturbations with an amplitude of 30%. The
amplitude and phase of the seasonal cycle return to more correct values. The
sensitivity experiments in Finck et al. (2013) verified the above-mentioned
issue.
### 4. Summary and discussions
The above findings can be summarized variable by variable:





-    **SLA:** In the period 1993-2013, the RMSD of daily SLA in the reanalysis
is gradually decreased from near 8 cm to less than 6 cm in the Pan-
Arctic region. The introduction of a bias estimation scheme proves very
efficient in constraining the bias. The largest RMSDs over 10 cm are
found in the Lofoten Basin. There is also a patch of larger error near
the ice edge, but observations are also less accurate there. There is a
weak seasonality in the performance of the system with the best in
summer months.

-    **SST:** The SST RMSD is about 0.66 °C, and the bias is about 0.04 °C.
Some biases along the sea-ice edge can be related to errors in the
observations. Other biases and RMSD relate to errors in the circulation
within the Nordic Sea. The transition to OSTIA high-resolution
observation is highly beneficial for constraining the bias and the RMSD,
but an overestimation of the observation error from the provider was
needed to avoid a collapse of the ensemble spread. The performance
of the system varies seasonally following the amount of observations
and a larger bias during summer months.

-    **Temperature and salinity profiles:** The errors are lower than the
WOA13 climatology near the surface (in the top of 100 to 200 m).
Below this depth, the model shows large biases and performs poorer
(RMSD > 1°C and about 0.1 psu). Some of the biases relate to the
limitations of the model to maintain the Atlantic water (as expected
from Ilicak et al. 2016) and others relate to a degradation introduced by
data assimilation (a flat multiplicative inflation). A large improvement
occurs at the time when the inflation method was replaced and when
more numerous observations were available from the IPY.



- **Sea ice concentration and extent:** TOPAZ4 agrees well with the OSI-
SAF sea ice concentrations. On average, the RMSDs are lower than
5% and the biases close to zero. The errors are larger close to the ice
edge, and poorest in the Greenland Sea. The errors are related to
biases in the thermodynamics and dynamics of the sea-ice model. The
bias is largest during the summer season. The performance is stable
throughout the reanalysis.
- **Sea ice drift:** The averaged drift in TOPAZ4 shows comparable
patterns to independent observation from IAPB buoys with the classical
Beaufort Sea gyre and transpolar drift. However the center of the gyre
is slightly misplaced. The RMSD of drift speed in the reanalysis is
about 5.1 km d$^{-1}$, and has a fast bias by about 2.5 km d$^{-1}$. The monthly
time variability compares well, but TOPAZ4 has a too weak seasonal
cycle and shifted by two months. From 2011 onwards, the atmospheric
drag coefficient was adjusted and the ice drift speed agrees better with
observations after the change. Still, with RMSDs of 5 km d$^{-1}$ close to
the signal itself, improving the performance of ice drift appears as a
priority for future operational use.
- **Sea ice thickness:** TOPAZ4 shows some large biases (approximately
-1.1 m) compared to ice thickness from ICESat and IceBridge as well
as compare to ice draft data. The bias is largest north of Ellesmere
Island with bias up to 2 m. The spatial pattern and regression compare
reasonably well. The ice is too thin in the period 2001-2010 due to
excessive snow depths and the seasonal cycle is too small during that
time.



The results from the 23-years reanalysis show overall a good stability and
good agreements with observations. However assessing the system for such
a long period also reveals some limitations that are inherent to the data
assimilation method or due to model flaws. In the following, we illustrate the
possible reasons and the means to tackle these in the future version of
TOPAZ5 system.
•   The Atlantic Waters have a too diffuse signature. In order to improve
their advection, we will double the horizontal and vertical resolution (58
hybrid layers and 5 km horizontal resolution). The parameterization of
diapycnal mixing will be reduced under sea-ice as proposed in Morison
et al. (1985). We also foresee that increasing the resolution will
improve the circulation in the Nordic Seas and reduce the associated
biases of SST and SSH.
•   The system has a too sharp ice edge. The current thermodynamic
model does not account for the heat capacity of the sea-ice. TOPAZ5
will transit to the community sea-ice mode CICE (Hunke et al. 2010),
which uses an improved thermodynamic parameterization.
•   Observations detect melt ponds as open water, whereas melt ponds
are not simulated in the current TOPAZ4. This creates bias in sea-ice
during summer months that is transferred to the interior of the ocean
via coupled data assimilation. In the future, we will choose the best
alternative between using an existing melt pond model or detect and
remove the signature of the melt ponds from the observations.
•   Comparisons against sea-ice drift and ice thickness highlighted more
severe limitations: Too thin ice, a too smooth thickness gradient from
Greenland into the Beaufort Gyre; the center of the Beaufort Gyre



being slightly misplaced, the sea-ice drift being too fast. These biases
can be reduced by optimizing the sea ice strength (P*) and the
atmospheric and ocean drag parameters (Massonnet et al. 2014).
However, optimal values of these parameters are moving targets in
view of their limited physical realism. The methodology proposed by
Barth et al. (2015), to estimate biases in atmospheric wind from ice drift
will also be considered. But the RMSDs of ice drift are relatively high (5
km d$^{-1}$ for an ice drift generally inferior to 10 km d$^{-1}$) although
comparable to short-term forecasts in Schweiger and Zhang (2015).
These fluctuating errors are less likely to be reduced by model tuning.
• There are further indications that the viscous-plastic and the related
elastic-viscous-plastic rheologies have inherent limitations for
simulating long-term properties of the ice drift – e.g. the acceleration of
sea ice drift, the phase of its seasonal cycle (Rampal et al. 2011). A
high-priority objective is therefore to couple TOPAZ to the neXtSIM
sea-ice model that is based on an elasto-brittle rheology. Recent
studies with forced version of neXtSIM (Bouillon and Rampal, 2015;
Rampal et al., 2016) suggest that the model is capable of reproducing
the sea ice deformations over a wide range of spatial and temporal
scales and reduces the error of the sea ice drift. It is of interest to
understand to which extent the coupling feedback will respond to this
improved dynamical model.
• The online bias estimation appeared quite successful to limit bias in our
model, but its implementation in the EnKF was very sensitive to the
choice of inflation method used. The present configuration that
combined r-factor inflation and autoregressive additive inflation is our



recommendation in a realistic system with a highly spatially variable observation network.

- The EnKF has proven capable to assimilate a large variety of observations, but more observations should be added. The sea-ice thickness of thin ice from the European Space Agency's (ESA) Soil Moisture and Ocean Salinity (SMOS) in Kaleschke et al. (2012) and Tian-Kunze et al. (2014). Also the complementary thickness of thick ice from ICESat (Kwok et al. 2009) and CryoSat-2 (Wingham et al., 2006; Laxon et al., 2013), and SMOS sea surface salinity (Reul et al., 2012).

**Acknowledgements**

Thanks to Dr. P. Rampal for processing the buoy dataset for sea ice drift and useful discussions. This study was supported by successive MyOcean projects from the European Commission (Grant numbers 218812), the Arctic element of the Copernicus Marine Services and a grant of CPU time from the Norwegian Supercomputing Project (NOTUR II grant number nn2993k).

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




Table 1. Overview of assimilated observations per cycle, average numbers for the cycles during which the observations are present. [1] The resolution of ice concentration product increased to 10 km. Unless specified, all observations from http://marine.copernicus.eu

| Type | Number | After SO | Spacing | Resolution | Period | Provider |
|---|---|---|---|---|---|---|
| SLA | $9\times10^4$ | $4\times10^4$ | Track | 7 km | 1992-2013 | CLS |
| SST | $6\times10^3$ | $6\times10^3$ | Gridded | 100 km | 1990-1998 | Reynolds SST from NCDC (http://www.nhc.noaa.gov/aboutsst.shtml) |
| SST | $2\times10^6$ | $2.4\times10^5$ | Gridded | 5 km | 1998-2013 | OSTIA from UK Met Office |
| In-situ T/S | $3\times10^4$ | $5\times10^3$ | Point | - | 1990-2013 | Ifremer + other |
| ICEC (SSM/I) | $9\times10^4$ | $5\times10^4$ | Gridded | 25 km | 1990-2002 | OSISAF |
| ICEC (AMSR-E) | $1.6\times10^5$ | $5\times10^4$ | Gridded | 12.5 km[1] | 2002-2013 | OSISAF |
| ICEC (AMSR-E) | $1.6\times10^5$ | $5\times10^4$ | Gridded | 12.5 km | 2008-2009 | AMSR-E (http://nsidc.org/data/amsre/) |
| Ice drift (CERSAT) | $5\times10^3$ | $5\times10^3$ | Gridded | 62.5 km | 2002-2010 | Ifremer |
| Ice drift (OSISAF) | $5\times10^3$ | $5\times10^3$ | Gridded | 62.5 km | 2011-2013 | OSISAF |
| Total | $2.3\times10^6$ | $4\times10^5$ | | | | |





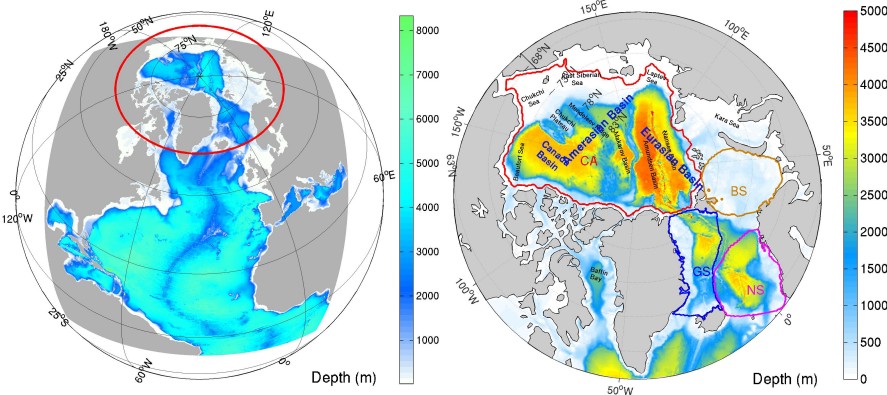

**Fig** 1. **Left**: Bottom topography in the whole TOPAZ4 domain. The red line delimits the Pan-Arctic
region north of 63°N. **Right**: Definition of sub-basins and marginal seas. The domain is divided
into the four sub-regions delimits by the colored lines: the Central Arctic in red (CA), the
Greenland Sea in blue (GS), the Barents Sea in orange (BS), and the Norwegian Sea in magenta
(NS).





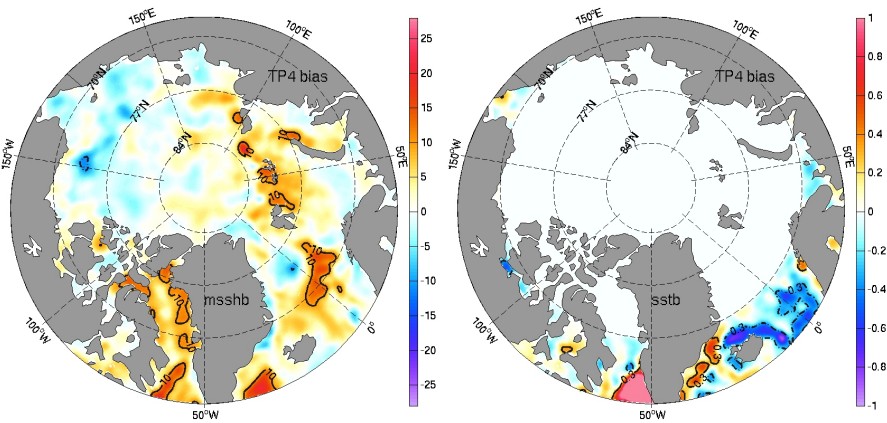

**Fig** 2. Estimates of the mean SSH bias (**Left**) and the SST bias (**Right**) obtained at last analyzed date
by online parameter estimation. In the left panel, the solid (dashed) line indicates the 10 (-10) cm
isolines. In the right panel, the solid (dashed) line indicates the 0.3 °C (-0.3 °C) isolines. There is
no bias estimation for SST in the white area north of 70°N.





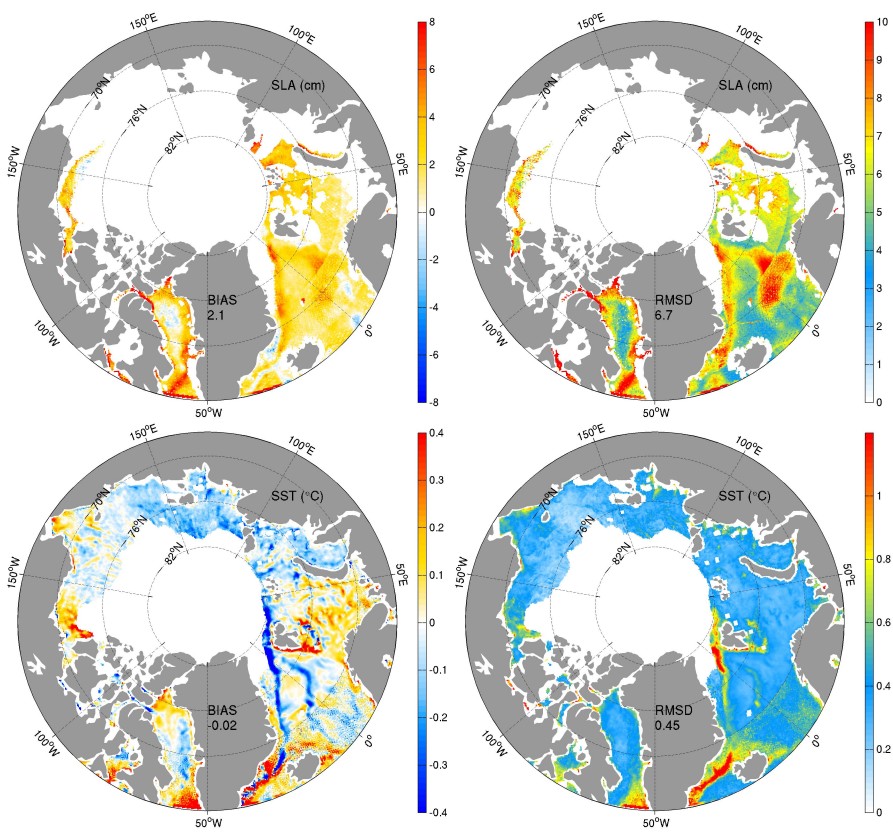

**Fig** 3. **Top**: Residual bias (left) and RMSD (right) between the daily average SLA from the reanalysis and the assimilated along-track SLA data averaged over the period 1993-2013 (unit: cm). **Bottom**: The corresponding residual bias (left) and RMSD (right) between the daily average SST from the reanalysis and the assimilated observations averaged over the period 1999-2013 (unit: °C). Areas with less than 30 observations have been masked in white.

10
11
12



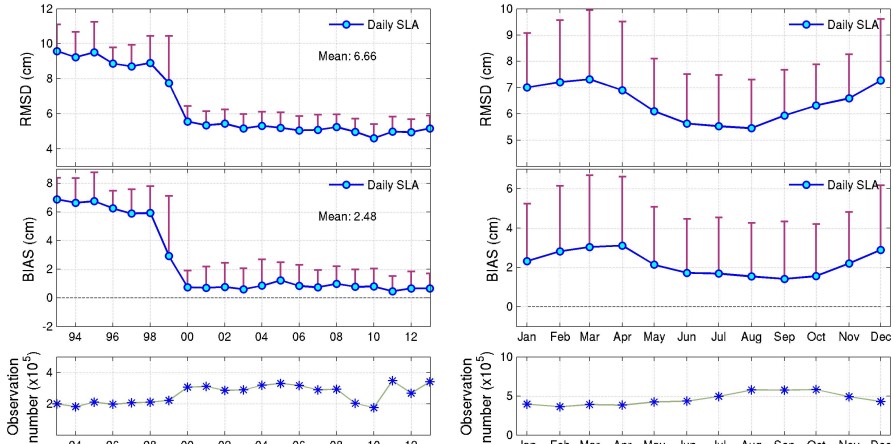

**Fig** 4. **Left**: Yearly averaged estimates of daily SLA RMSD (upper) and the residual bias (middle) of
the TOPAZ reanalysis calculated against the along-track SLA available in the Pan-Arctic region
(unit: cm). The error bars denote the standard deviations of the daily statistics within each year.
The bottom panel is the number of available observations in each year. **Right**: Similar plot for
monthly averaged estimate of daily SLA RMSD (upper), and the residual bias (middle). The error
bars denote the standard deviations of the daily statistic within each month. The bottom panel
shows the number of observations available for each month in the Pan-Arctic during 1993-2013.





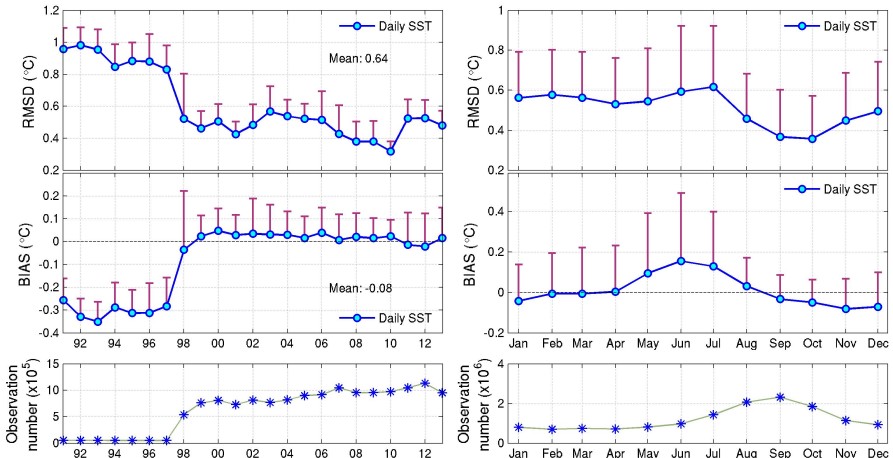

**Fig** 5. Same as the previous figure but for SST for the period 1991-2013 (unit: °C).





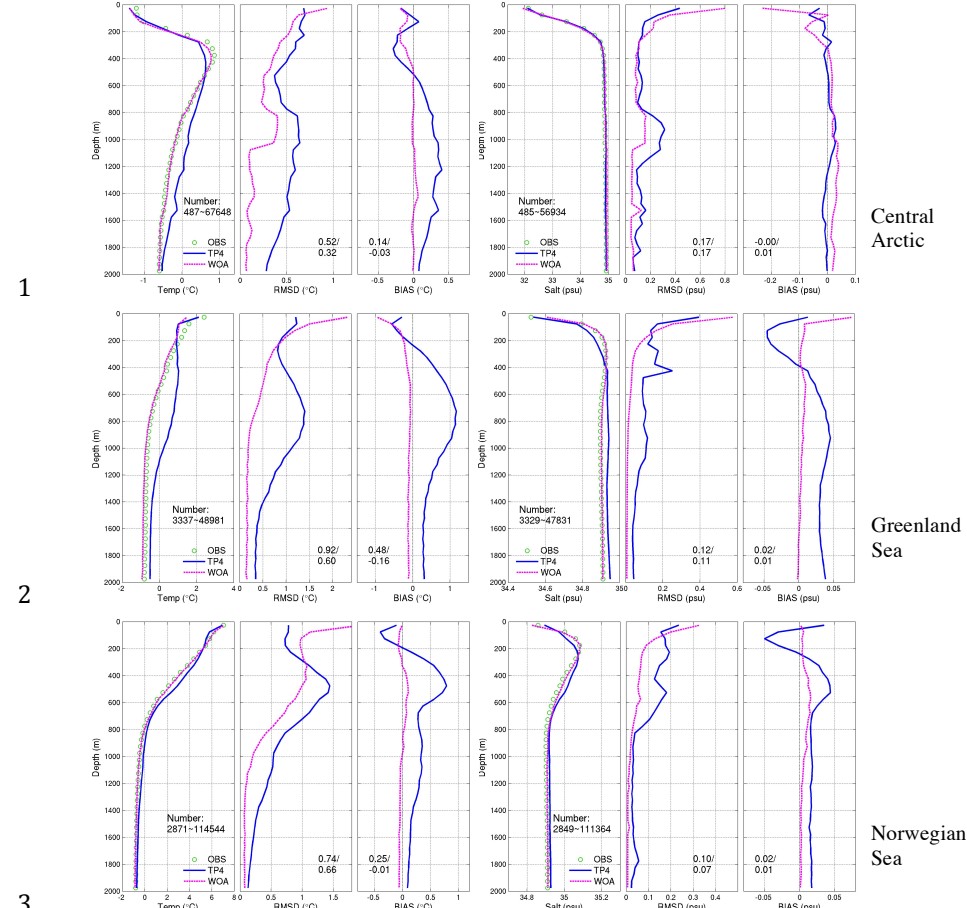





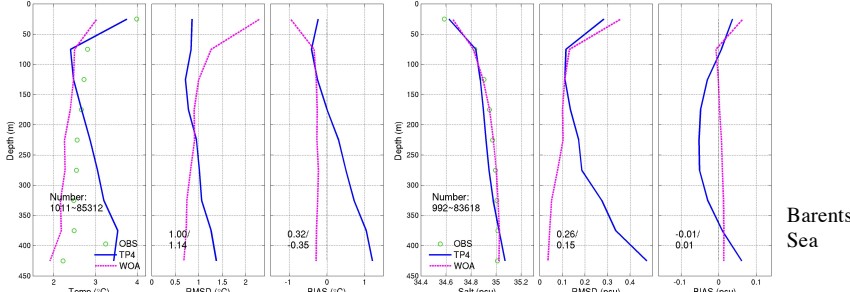

**Fig** 6. The mean profiles of temperature (*left*) and salinity (*right*) and the corresponding bias and
RMSD in each of the marginal seas of the Pan-Arctic region. The green circle is the observations,
the blue lines are the TOPAZ reanalysis, and the pink lines are from the WOA13 climatology.
The numbers in the first-column subpanels are the minimal and maximal number of observations
available in each of 50 m depths; the upper numbers in the other-column subpanels are the mean
estimate in vertical for TOPAZ reanalysis, and the lower numbers is for WOA13.




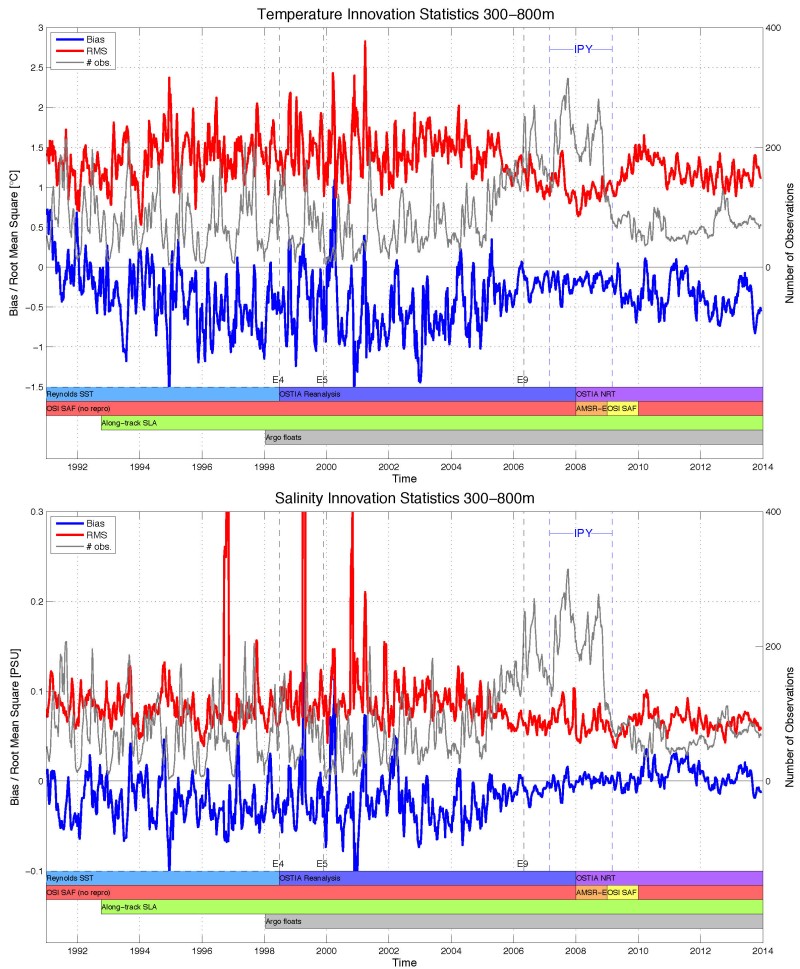

**Fig** 7. Time series of innovation statistics for temperature (**top**) and salinity (**bottom**) observed at the depth of between 300-800 m depths. The bias is plotted with a blue line, the RMSD is in red and the number of assimilated observations is plotted with a grey line. The time series are filtered with a 28 days moving window. The vertical dashed lines indicate the major change events in the course of the TOPAZ reanalysis.



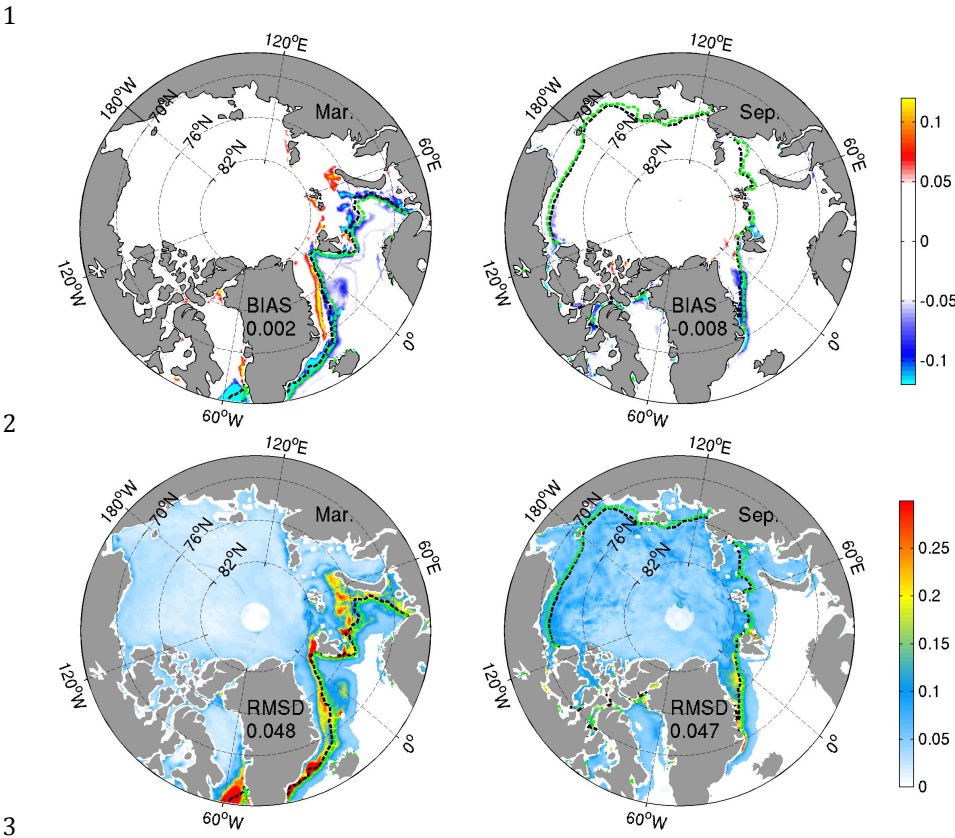

**Fig** 8. Spatial bias (upper) and RMSD (lower) of sea ice concentration in the TOPAZ reanalysis for
March (*left*) and September (*right*) calculated from the daily averages for the period 1991-2013.
The dashed black (green) lines delimit the monthly mean sea ice edges (at 15%) in the TOPAZ
reanalysis (OSISAF).



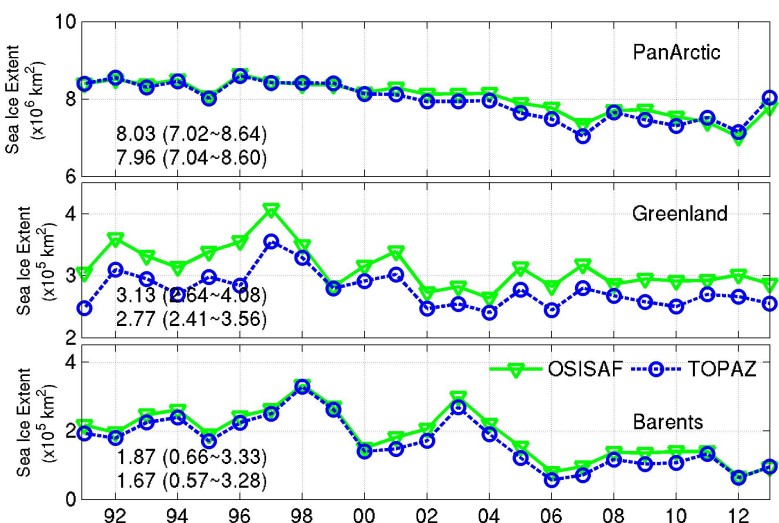

**Fig** 9. Yearly time series of the sea ice extent in the Pan-Arctic region, the Greenland Sea, and the Barents Sea from TOPAZ reanalysis (dashed) and OSISAF (solid).




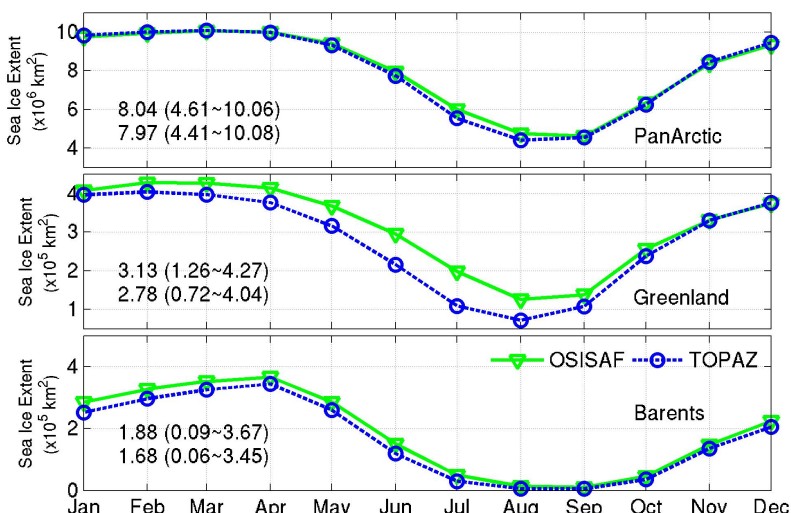

5  **Fig** 10. Seasonality of the sea ice extents in the TOPAZ reanalysis (blue line) and OSISAF (green line)
6      in the Pan-Arctic Ocean, Greenland Sea, and Barents Sea regions.





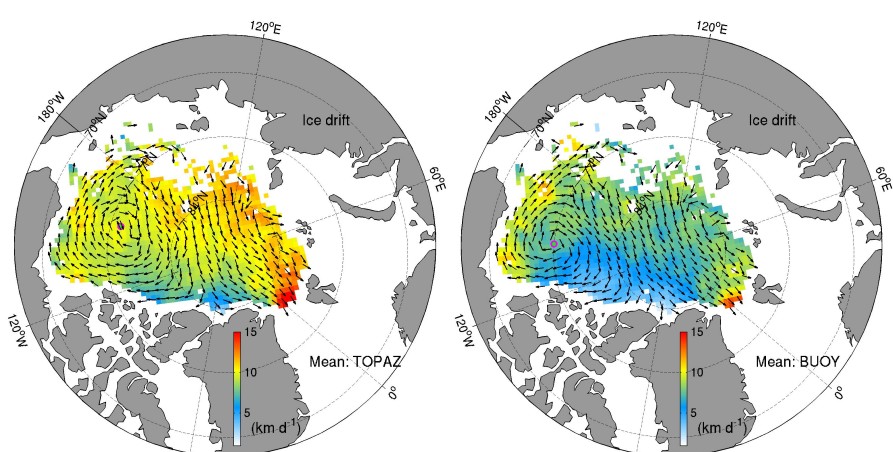

**Fig** 11. Sea ice drift vectors (*arrows*) and speeds (*color shading*) averaged over the period 1991-2011
for (a) TOPAZ reanalysis and (b) IABP buoys. The center of the anticyclonic Beaufort Gyre is
marked with a magenta circle at (155°W, 78.1°N) in the TOPAZ reanalysis and (145°W, 77°N) in
the observations respectively.





**Fig** 12. Monthly time series of the daily averaged sea ice drift speeds in the Central Arctic from the
TOPAZ reanalysis (blue line) and the IABP buoys (green line) during 1991-2011. The error bars
represent the standard deviations of the daily estimates for each month.






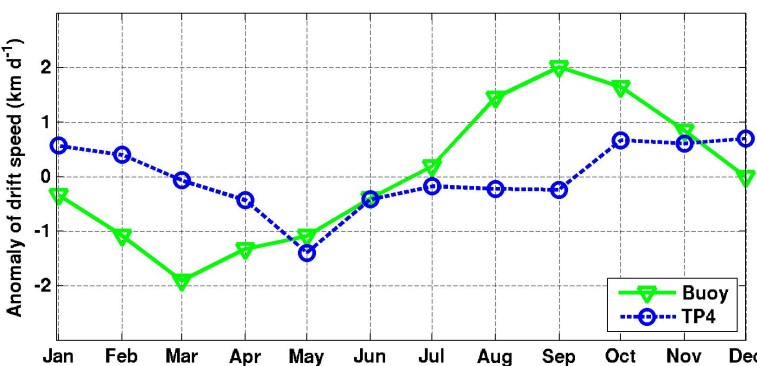

Fig 13. Seasonality of the sea ice drift velocities from the reanalysis and the buoy during 1991-2011.




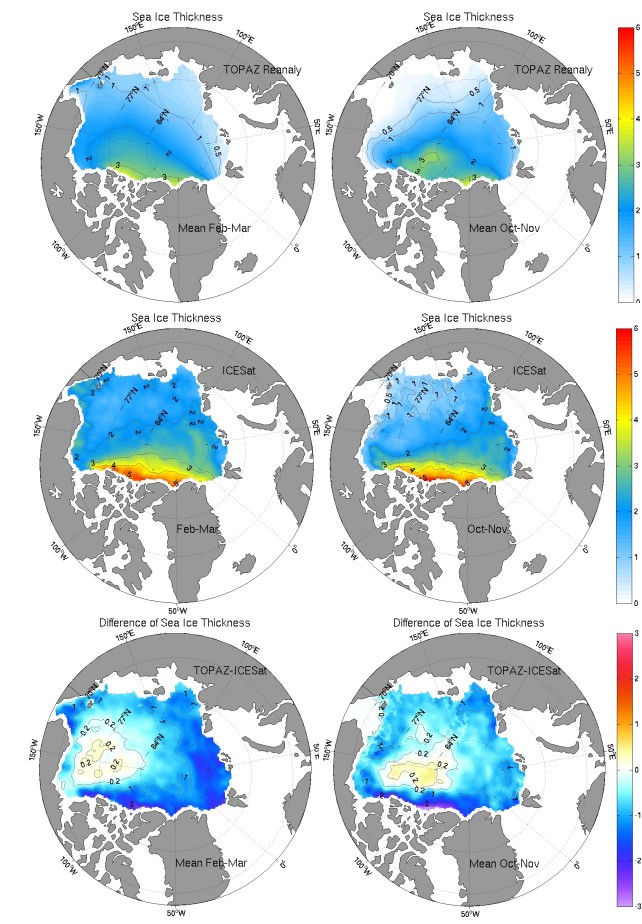

**Fig** 14. Mean sea ice thicknesses from TOPAZ (upper) and ICESat (middle), and their difference
(bottom) for February-March (*in left column*) and October-November (*in right column*) averaged
over the period 2003-2008.






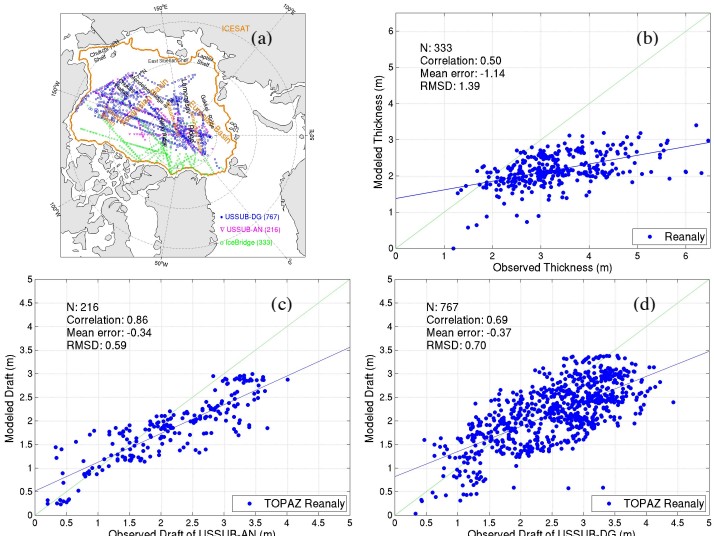

**Fig** 15. Validation the sea ice thickness in the TOPAZ reanalysis versus available in situ observations. (a) Locations of in situ observations available from IceBridge, USSUB-AN and USSUB-DG in the Central Arctic. Regression analysis of TOPAZ reanalysis (b) vs. IceBridge; (c) vs. USSUB-AN; (d) vs. USSUB-DG.





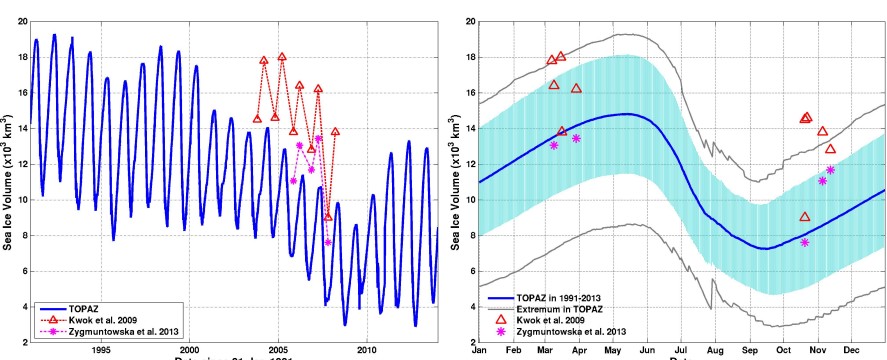

**Fig** 16. **Left**: Time series of the daily averaged sea ice volume in the Arctic from the TOPAZ4 (blue
line) and the observations from Kwok et al. (2009) and from Zygmuntowska et al. (2013). **Right**:
Daily time series of the averaged sea ice volume in the Arctic from the TOPAZ4 for the period
1991-2013 (blue line) and the standard deviation shown as the cyan error-bar. The gray lines
represent the extreme volumes in the 23 years. The triangle and start markers are the observations
estimated by Kwok et al. (2009) and Zygmuntowska et al. (2013) respectively.

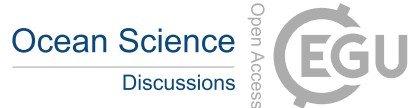


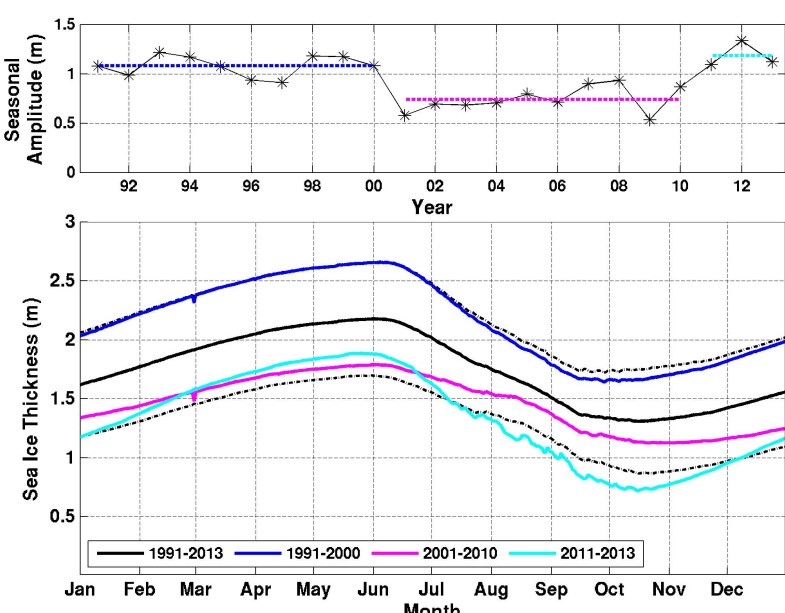

**Fig** 17. **TOP**: Yearly time series of the seasonal amplitudes of the mean sea ice thickness in the Central
Arctic with the solid black line. The dashed lines represent the averaged estimate for: 1991-2000,
2001-2010, and 2011-2013 (1.08, 0.74, and 1.18 m respectively). **Bottom**: Daily time series of the
mean sea ice thickness in the Central Arctic for three different time periods. The black dashed
lines denote the standard deviation for the 23 yearly estimates.

