# Peer review of "Quality assessment of the TOPAZ4 reanalysis in"

_Ocean Science, 2016_

## Referee Comment (RC1) · Anonymous Referee #1 · 26 Jul 2016

This paper describes the result of a 23-year reanalysis (1991-2013) of the Arctic, as obtained from the TOPAZ4 coupled ocean and sea ice data assimilation system. The paper mainly provides a detailed comparison between the reanalysis and available observation datasets (sea level anomaly, sea surface temperature, in situ temperature and salinity profiles, sea ice concentration, sea ice drift and sea ice thickness). In its present form, the paper is essentially descriptive; it does not provide really new scientific ideas; and the method used to assess the ensemble reanalysis (statistics of the difference between ensemble mean and observations) is quite crude and not very original. Nevertheless, as I understand, this paper is meant to be the reference paper to an important new reanalysis product describing the Arctic Ocean (delivered by the Arctic component of the MyOcean system). As such, I think that this paper could deserve publication in Ocean Science.

I have however a few concerns concerning the manuscript that I believe should be taken into account in a revised version.

1. The main purpose of the paper is the assessment of the reanalysis using all available observations. However, to compare the reanalysis to observations, the authors just compute the average and RMS difference between the ensemble mean and observations. This method looks very crude to me, and does not meke justice to the advanced method that is used to perform data assimilation. The ensemble data assimilation system provides a probability distribution for the reanalysis, which is described by an ensemble of model states. Why then assessing the reanalysis using the ensemble mean only? Probabilistic tools exist to perform an objective comparison between ensemble simulations and observations (see for instance Toth et al., 2003, or Candille et al., 2007). Why performing an ensemble reanalysis if the probabilistic information is dicarded to study the performance of the system? Would it be possible to include some kind of probabilistic assessment, or at least explain better why using such a crude assessment method?

2. In assessing the performance by computing the difference with observations, the paper implicitly (and sometimes explicitly) assumes that the closer to the observations, the better the reanalysis. This amount to completely neglecting observation errors in the assessment of the reanalysis, which is usually not an appropriate approximation. This incorrect assumption is for instance made explicitly in:

   - p. 13, l. 4, where the misfit to observations is called "error" on the reanalysis;
   - p. 14, l. 21, where the reanalysis is said to be improved if difference to observations is smaller;
   - p. 16, l. 6-7, where it is said that an RMSD with observations of 5% is good

whereas the accuracy of the observations is said to be about 10%. In my view, this just mean that the reanalysis is excessively close to observations.

I think that it would be important to better explain the limitations of this simple approach for assessing the performance of the reanalysis; to explain why more sophisticated comparison metrics were not applied (see my previous comment) and avoid the misleading expressions listed above.

3. In the introduction, the authors provide several arguments to support the idea that ensemble methods are an appropriate way to apply the dynamical model constraint in the estimation process. However, this is not discussed anymore in the assesment of the performance of the reanalysis. Only quantitative difference to observations are provided and analysed. I think that the quality of the paper would be enhanced if more explicit evidence of what is stated in the introduction was provided in addition to the simple description of the distance between reanalysis and observations.

**References:**

Candille, G., Côté, C., Houtekamer, P. L., and Pellerin, G.: Verification of an ensemble prediction system against observations, Mon. Weather Rev., 135, 2688–2699, 2007.

Toth, Z., Talagrand, O., Candille, G., and Zhu, Y.: Probability and ensemble forecasts, in: Forecast Verification: a Practitioner's Guide in Atmospheric Science, edited by: Jolliffe, I. and Stephenson, D. B., Wiley, UK, 137–163, 2003.

---

## Referee Comment (RC2) · Anonymous Referee #2 · 7 Oct 2016

The manuscript provides a detailed description of the results of the 23-year reanalysis of the Arctic computed with the TOPAZ4 model system.

The manuscript appears more like a report than a scientific paper tackling a scientific or methodological issue. The model system is described elsewhere and has undergone very little changes with respect to previously published information. The assessment of the quality of the products uses a rather elementary approach.

The results discussed in the manuscript can be useful as a support of further studies using the reanalysed fields but, as it stands, the manuscript is merely descriptive. Also, little information is given about the ensemble and this information is not used to asses the quality of the reanalysis : only the ensemble mean are used for this purpose.

The quality of the reanalysis obtained using TOPAZ4 could also be compared with the

quality of similar other products.

---

## Author Comment (AC1) · 4 Nov 2016

Referee #1 This paper describes the result of a 23-year reanalysis (1991-2013) of the Arctic, as obtained from the TOPAZ4 coupled ocean and sea ice data assimilation system. The paper mainly provides a detailed comparison between the reanalysis and available observation datasets (sea level anomaly, sea surface temperature, in situ temperature and salinity profiles, sea ice concentration, sea ice drift and sea ice thickness). In its present form, the paper is essentially descriptive; it does not provide really new scientific ideas; and the method used to assess the ensemble reanalysis (statistics of the difference between ensemble mean and observations) is quite crude and not very original. Nevertheless, as I understand, this paper is meant to be the reference paper to an important new reanalysis product describing the Arctic Ocean (delivered by the Arctic component of the MyOcean system). As such, I think that this paper could

deserve publication in Ocean Science. I have however a few concerns concerning the manuscript that I believe should be taken into account in a revised version. 1. The main purpose of the paper is the assessment of the reanalysis using all available observations. However, to compare the reanalysis to observations, the authors just compute the average and RMS difference between the ensemble mean and observations. This method looks very crude to me, and does not make justice to the advanced method that is used to perform data assimilation. The ensemble data assimilation system provides a probability distribution for the reanalysis, which is described by an ensemble of model states. Why then assessing the reanalysis using the ensemble mean only? Probabilistic tools exist to perform an objective comparison between ensemble simulations and observations (see for instance Toth et al., 2003, or Candille et al., 2007). Why performing an ensemble reanalysis if the probabilistic information is discarded to study the performance of the system? Would it be possible to include some kind of probabilistic assessment, or at least explain better why using such a crude assessment method? Would it be possible to include some kind of probabilistic assessment, or at least explain better why using such a crude assessment method?

Reply: We would like to thank the reviewer for this constructive comment and suggestion. Our main purpose is to present and validate the official product of Copernicus CMEMS for the Arctic region, which is provided as a deterministic reanalysis product based on the ensemble mean, for consistency with other CMEMS reanalyses. However, we fully agree that validation of the quality of the ensemble is crucial to prove the ability of the reanalysis to make the best use of a heterogeneous observational network (spatially, temporally and various data sources); for example that we do not overfit one observational data set at the expense of the others. The reliability of a system is important as well for an EnKF-based data assimilation system like ours, since the efficiency of the system relies on adequate assumptions for model and observation errors. Unfortunately, our storage facility is insufficient to store the full ensemble of the daily averaged fields, and we only have at our disposal the ensemble statistics of the variables assimilated at each assimilation time (every week). In order to address the

reviewer comment, we propose to extend our validation work with a reliability analysis (e.g. Candille et al. 2007, Desroziers et al. 2005, Rodwell et al., 2016) of the observation network assimilated (SST, SSH, Ice concentration, T-S). This metric will be used to assess the behaviour of our assimilation system in space and in time.

2. In assessing the performance by computing the difference with observations, the paper implicitly (and sometimes explicitly) assumes that the closer to the observations, the better the reanalysis. This amount to completetly neglecting observation errors in the assessment of the reanalysis, which is usually not an appropriate approximation. This incorrect assumption is for instance made explicitly in: • p. 13, l. 4, where the misfit to observations is called "error" on the reanalysis; • p. 14, l. 21, where the reanalysis is said to be improved if difference to observations is smaller; • p. 16, l. 6-7, where it is said that an RMSD with observations of 5% is good; whereas the accuracy of the observations is said to be about 10%. In my view, this just mean that the reanalysis is excessively close to observations. I think that it would be important to better explain the limitations of this simple approach for assessing the performance of the reanalysis; to explain why more sophisticated comparison metrics were not applied (see my previous comment) and avoid the misleading expressions listed above.

Reply: We agree with the reviewer and the above statements will be revised according to the reliability analysis. We will add when possible the quantity sqrt(obs_error+ens_spread) to ensure that we are not over-fitting observations and that the ensemble does not collapse.

3. In the introduction, the authors provide several arguments to support the idea that ensemble methods are an appropriate way to apply the dynamical model constraint in the estimation process. However, this is not discussed anymore in the assessment of the performance of the reanalysis. Only quantitative difference to observations are provided and analysed. I think that the quality of the paper would be enhanced if more explicit evidence of what is stated in the introduction was provided in addition to the simple description of the distance between reanalysis and observations.

Reply: The overall text will be revised according the change proposed above.

---

## Author Comment (AC2) · 4 Nov 2016

Referee #2 The manuscript provides a detailed description of the results of the 23-year reanalysis of the Arctic computed with the TOPAZ4 model system. 1. The manuscript appears more like a report than a scientific paper tackling a scientific or methodological issue. The model system is described elsewhere and has undergone very little changes with respect to previously published information. The assessment of the quality of the products uses a rather elementary approach.

Reply: The paper by Sakov et al. (2012) was a proof of concept that an EnKF-based assimilation system can be used with a coupled ocean and sea ice for long reanalysis. This study does not propose new methodological development but it verifies that the proof of concept holds when applied for a longer period (23 years are more relevant

to the community than 6 years) with a more heterogeneous observation network (spatially, temporally and various data sources). The main purpose of the manuscript is to present and validate the official Copernicus CMEMS product for the Arctic region. The proposed reanalysis is unique (see table below extracted from Chevalier et al. 2016) as it proposes a long high-resolution dynamical reconstruction of the ocean and sea ice, and assimilates a complete set of observations available in the Arctic region with an advanced ensemble data assimilation method and with strongly coupled data assimilation between ocean and sea-ice. We have tried to present this achievement in a concise manner, with a primary focus to inform the end-user about the strength and weaknesses of our data set. As a response to the recommendation of the first reviewer (and your following comment), we will extend the current validation with the an analysis of the ensemble reliability, and asses whether our system manage to provide a dynamical reconstruction that falls within the uncertainty of the different observational data sets that are assimilated. We believe that this will increase the scientific value of our manuscript as it would confirm or infirm the underlying assumptions on model and observation errors.

2. The results discussed in the manuscript can be useful as a support of further studies using the reanalysed fields but, as it stands, the manuscript is merely descriptive. Also, little information is given about the ensemble and this information is not used to assess the quality of the reanalysis: only the ensemble mean are used for this purpose.

Reply: We agree and this will be investigated using the ensemble statistics at assimilation time. See also our answer to the other reviewer.

3. The quality of the reanalysis obtained using TOPAZ4 could also be compared with the quality of similar other products.

Reply: We think that such comparison is beyond the scope of our paper and, for the sake of diplomatic correctness, is better undertaken in a separate collaborative initiative (The ongoing Ocean Synthesis COST action, a follow-up of the ORA-IP Arctic paper

by Chevallier et al.). A primary comparison of the ocean part of our analysis has been compared with other existing systems (Lien et al. 2016, cited in the manuscript).

[Figure]

Table 1  System configuration and selected parameters

| Name | C-GLORS05 | CNRM | ECCO-v4 | ECDA | GloSea5 | G2V3 | MERRA Ocean | MOVE-CORE | MOVE-G2 | ORAP5 | UR025.4 | G2V1 | ERAL | ERAN |
|---|---|---|---|---|---|---|---|---|---|---|---|---|---|---|
| Institution | CMCC | CNRM-GAME | JPL/NASA, MIT, AER | GFDL/NOAA | UK Met Office | Mercator Océan | GSFC/NASA/GMAO | MRI/JMA | MRI/JMA | ECMWF | University of Reading | Mercator Océan | ECMWF | ECMWF |
| Nominal horizontal resolution | 0.5° | 1° | 0.4°–1.0° | 1° | 0.25° | 0.25° | 0.5° | 0.5° × 1° | 0.3–0.5° × 1° | 0.25° | 0.25° | 0.25° | 1° | 1° |
| Ocean–sea ice model | NEMO3.2-LIM2 | NEMO3.2-GELATO5 | MITgcm | GFDL-MOM4.4.1-SIS | NEMO3.2-CICE4.0 | NEMO3.1-LIM2 (EVP) | MOM4.1-CICE4.0 | MRI.COM3-Mellor & Kanta + CICE4.0 | MRI.COM3-Mellor & Kanta + CICE4.0 | NEMO3.4-LIM2 | NEMO3.2-LIM2 | NEMO3.1-LIM2 (EVP) | NEMO3.2-LIM2 | NEMO3.2-LIM2 |
| Time period | 1979–2011 | 1990–2010 | 1992–2010 | 1961–2014 | 1993–2012 | 1993–2011 | 1979–present | 1948–2007 | 1993–2012 | 1979–2012 | 1989–2010 | 1993–2009 | 1990–2011 | 1990–2011 |
| Source of atmospheric forcing data | ERA-Interim | ERA-Interim | ERA-Interim | Coupled run constrained to NCEP/NCAR-NCEP/DOE | ERA-Interim | ERA-Interim | Coupled run constrained to MERRA | CORE | JRA55 | ERA-Interim | ERA-Interim | ERA-Interim | ERA-Interim | ERA-Interim |
| Vertical discretization | 2 ice + 1 snow | 9 ice + 1 snow | 1 ice + 1 snow | 2 ice + 1 snow | 1 ice + 1 snow | 2 ice + 1 snow | 4 ice + 1 snow | 1 ice + 1 snow | 1 ice + 1 snow | 2 ice + 1 snow | 2 ice + 1 snow | 2 ice + 1 snow | 2 ice + 1 snow | 2 ice + 1 snow |
| Thickness categories | 1 | 8 | 1 | 5 | 5 | 1 | 5 | 5 | 5 | 1 | 1 | 1 | 1 | 1 |
| Dynamics | EVP | EVP | VP | EVP | EVP | EVP | EVP | EVP | EVP | VP | VP | EVP | VP | VP |
| P* (N/m)/Cf (—) | $P* = 2.0 \times 10^4$ | $P* = 2.75 \times 10^4$ | $P* = 2.754 \times 10^4$ | $P* = 2.5 \times 10^4$ | $Cf = 17$ | $P* = 2 \times 10^4$ | $P* = 2.75 \times 10^4$ | $P* = 2.75 \times 10^4$ | $P* = 2.75 \times 10^4$ | $P* = 1.50 \times 10^4$ | $P* = 1$ | $P* = 2 \times 10^4$ | $P* = 1.5 \times 10^4$ | $P* = 1.5 \times 10^4$ |
| Drag air–ice (10-3) | 1.63 | 1.63 | 2.00 | 1.21 | 1.63 | 1.50 | 1.63 | 3.00 | 1.00 | 1.63 | 1.63 | 1.50 | 1.63 | 1.63 |
| Drag ocean–ice (10-3) | 10.00 | 5.00 | 1.00 | 3.24 | 5.36 | 10.00 | 5.36 | 5.50 | 5.50 | 10.00 | 5.00 | 10.00 | 5.00 | 5.00 |
| DA sea ice system | Linear nudging | None (SST) | Adjoint | None (SST) | 3DVAR | 2D local analysis SEEK filter | EnOI | None (SST) | None (SST) | 3DVAR-FGAT | OI | None (SST) | Linear nudging | Flow-dependent nudging |
| DA sea ice data | NSIDC | – | NSIDC | – | OSI-SAF | CERSAT | NSIDC | – | – | OSTIA | OSI-SAF | – | NCEP-OIv2 | NCEP-Oiv2 |
| Analysis window | 7 days | 10 days | 20 years | 1 day | 1 day | 7 days | 5 days | 1 month | 1/3 month | 5 days | 5 days | 7 days | 1 day | 1 day |

P* and Cf are parameters for the ice strength formulations following respectively Hibler (1979) and Rothrock (1975)

*DA* data assimilation, *VP* viscous-plastic, *EVP* elastic–viscous–plastic, *SST* sea surface temperature

M. Chevallier et al.

**Fig. 1.**

---

## Author Response (AR1)

**Responses to reviewers' comments**

We would like to thank once more the reviewers for their comments. The major changes in the revision are listed as following:

- Add Section 3 to assess the ensemble reliability in the TOPAZ4 reanalysis.
- Shortened the model and assimilation description to reduce overlap with Sakov et al. (2012).
- Improving the uncertainty analysis of the reanalysis with respect to in situ profiles in Section 4.
- Add figures 3, 4 and 8 into the revision
- Changes of the text in agreement with the recommendations of the reviewers.

The detailed responses are listed one by one with blue as following:

**Referee #1**

*1.       The main purpose of the paper is the assessment of the reanalysis using all available observations. However, to compare the reanalysis to observations, the authors just compute the average and RMS difference between the ensemble mean and observations. This method looks very crude to me, and does not make justice to the advanced method that is used to perform data assimilation. The ensemble data assimilation system provides a probability distribution for the reanalysis, which is described by an ensemble of model states. Why then assessing the reanalysis using the ensemble mean only? Probabilistic tools exist to perform an objective comparison between ensemble simulations and observations (see for instance Toth et al., 2003, or Candille et al., 2007). Why performing an ensemble reanalysis if the probabilistic information is discarded to study the performance of the system? Would it be possible to include some kind of probabilistic assessment, or at least explain better why using such a crude assessment method? Would it be possible to include some kind of probabilistic assessment, or at least explain better why using such a crude assessment method?*

**Reply:** We would like to thank the reviewer for this constructive comment and suggestion. Our main purpose is to present and validate the official product of Copernicus CMEMS for the Arctic region, which is provided as a deterministic reanalysis product based on the ensemble mean, for consistency with other CMEMS reanalyses. However, we fully agree that validation of the quality of the ensemble is crucial to prove the ability of our reanalysis to make the best use of the heterogeneous observational network (spatially, temporally and various data sources); for example that we do not overfit one observational data set at the expense of the others. The reliability for an EnKF-based data assimilation system like ours is even rather important, since the efficiency of the system relies on adequate assumptions for model and observation errors. Unfortunately, our storage facility is insufficient to store the

full ensemble of the daily averaged fields, and we only have at our disposal the ensemble statistics of the variables assimilated at each assimilation time (every week).

In order to address the reviewer comment, we have extended our validation work with a reliability analysis (e.g. Candille et al. 2007) of the observation network assimilated according to the all assimilated variables (SST, SSH, Ice concentration, T-S, and sea ice drift).

*2. In assessing the performance by computing the difference with observations, the paper implicitly (and sometimes explicitly) assumes that the closer to the observations, the better the reanalysis. This amount to completely neglecting observation errors in the assessment of the reanalysis, which is usually not an appropriate approximation. This incorrect assumption is for instance made explicitly in:*

- *p. 13, l. 4, where the misfit to observations is called "error" on the reanalysis;*

**Reply:** Thanks. It is corrected with "of the misfit".

- *p. 14, l. 21, where the reanalysis is said to be improved if difference to observations is smaller;*

**Reply:** Thanks. We have noted that the increased accuracy is an improvement because the reliability remained equal in the meantime.

- *p. 16, l. 6-7, where it is said that an RMSD with observations of 5% is good; whereas the accuracy of the observations is said to be about 10%. In my view, this just mean that the reanalysis is excessively close to observations.*

**Reply:** Thank you, it is corrected. The reliability analysis in Section 3 revealed - in the contrary - an underdispersion. The sentence now concentrates on the qualitative message (errors concentrated near the ice edge).

*I think that it would be important to better explain the limitations of this simple approach for assessing the performance of the reanalysis; to explain why more sophisticated comparison metrics were not applied (see my previous comment) and avoid the misleading expressions listed above.*

**Reply**: We agree with the reviewer and the above statements will be revised according to the reliability analysis. We added the reliability analyses of the modified RCRV to ensure that we are not over-fitting observations and that the ensemble does not collapse, and also use the innovation budget (Rodwell et al., 2016) to investigate the uncertainty variability in time.

*3. In the introduction, the authors provide several arguments to support the idea that ensemble methods are an appropriate way to apply the dynamical model constraint in the estimation process. However, this is not discussed anymore in the assessment of the performance of the reanalysis. Only quantitative difference to observations are provided and analysed. I think that the quality of the paper would be enhanced if more explicit evidence of what is stated in the introduction was provided in addition to the simple description of the distance between reanalysis and observations.*

**Reply**:  This is now extensively discussed in the manuscript, both during the reliability section and in the conclusion. Despite some discontinuities causes by the change of observational data set and change in the data assimilation setting, the statistic remains relatively stable through the course of the reanalysis. The reliability budget analysis exemplifies the challenge of providing a balances reanalysis with relative contributions from various data sources.

**Referee #2**

*The manuscript appears more like a report than a scientific paper tackling a scientific or methodological issue. The model system is described elsewhere and has undergone very little changes with respect to previously published information. The assessment of the quality of the products uses a rather elementary approach.*

**Reply**: The paper by Sakov et al. (2012) was a proof of concept that an EnKF-based assimilation system can be used with a coupled ocean and sea ice for long reanalysis. This study does not propose new methodological development but it verifies that the proof of concept holds when applied for a longer period (23 years are more relevant to the community than 6 years) with a more heterogeneous observation network (spatially, temporally and various data sources). The main purpose of the manuscript is to present and validate the official Copernicus CMEMS product for the Arctic region. The proposed reanalysis is unique (see table below extracted from Chevalier et al. 2016) as it proposes a long high-resolution dynamical reconstruction of the ocean and sea ice, and assimilates a complete set of observations available in the Arctic region with an advanced ensemble data assimilation method and with strongly coupled data assimilation between ocean and sea-ice. We have tried to present this achievement in a concise manner, with a primary focus to inform the end-user about the strength and weaknesses of our data set. As a response to the recommendation of the first reviewer (and your following comment), we have extended the current

validation with the analysis of the ensemble reliability, and asses whether our system manage to provide a dynamical reconstruction that falls within the uncertainty of the different observational data sets that are assimilated. We believe it has increased the scientific value of our manuscript.

**Table 1** System configuration and selected parameters

| Name | C-GLORS05 | CNRM | ECCO-v4 | ECDA | GloSea5 | G2V3 | MERRA Ocean | MOVE-CORE | MOVE-G2 | ORAP5 | UR025.4 | G2V1 | ERA L | ER AN |
|---|---|---|---|---|---|---|---|---|---|---|---|---|---|---|
| Institution | CMCC | CNRM-GAME | JPL/NASA, MIT, AER | GFDL/NOAA | UK Met Office | Mercator Océan | GSFC/NASA/GMAO | MRI/JMA | MRI/JMA | ECMWF | University of Reading | Mercator Océan | ECMWF | ECMWF |
| Nominal horizontal resolution | 0.5° | 1° | 0.4°–1.0° | 1° | 0.25° | 0.25° | 0.5° | 0.5° × 1° | 0.3–0.5° × 1° | 0.25° | 0.25° | 0.25° | 1° | 1° |
| Ocean–sea ice model | NEMO3.2-LIM2 | NEMO3.2-GELATO5 | MITgcm | GFDL-MOM4.4.1-SIS | NEMO3.2-CICE4.0 | NEMO3.1-LIM2 (EVP) | MOM4.1-CICE4.0 | MRI COM3-Mellor & Kanta + CICE4.0 | MRI.COM3-Mellor & Kanta + CICE4.0 | NEMO3.4-LIM2 | NEMO3.2-LIM2 | NEMO3.1-LIM2 (EVP) | NEMO3.2-LIM2 | NEMO3.2-LIM2 |
| Time period | 1979–2011 | 1990–2010 | 1992–2010 | 1961–2014 | 1993–2012 | 1993–2011 | 1979–present | 1948–2007 | 1993–2012 | 1979–2012 | 1989–2010 | 1993–2009 | 1990–2011 | 1990–2011 |
| Source of atmospheric forcing data | ERA-Interim | ERA-Interim | ERA-Interim | Coupled run constrained to NCEP/NCAR-NCEP/DOE | ERA-Interim | ERA-Interim | Coupled run constrained to MERRA | CORE | JRA55 | ERA-Interim | ERA-Interim | ERA-Interim | ERA-Interim | ERA-Interim |
| Vertical discretization | 2 ice + 1 snow | 9 ice + 1 snow | 1 ice + 1 snow | 2 ice + 1 snow | 1 ice + 1 snow | 2 ice + 1 snow | 4 ice + 1 snow | 1 ice + 1 snow | 1 ice + 1 snow | 2 ice + 1 snow | 2 ice + 1 snow | 2 ice + 1 snow | 2 ice + 1 snow | 2 ice + 1 snow |
| Thickness categories | 1 | 8 | 1 | 5 | 5 | 1 | 5 | 5 | 5 | 1 | 1 | 1 | 1 | 1 |
| Dynamics | EVP | EVP | VP | EVP | EVP | EVP | EVP | EVP | EVP | VP | VP | EVP | VP | VP |
| P* (N/m)/ Cf (–) | P* = $2.0 \times 10^4$ | P* = $2.75 \times 10^4$ | P* = $2.754 \times 10^4$ | P* = $2.5 \times 10^4$ | Cf = 17 | P* = $2 \times 10^4$ | P* = $2.75 \times 10^4$ | P* = $2.75 \times 10^4$ | P* = $2.75 \times 10^4$ | P* = $1.50 \times 10^4$ | P* = $1 \times 10^4$ | P* = $2 \times 10^4$ | P* = $1.5 \times 10^4$ | P* = $1.5 \times 10^4$ |
| Drag air–ice (10-3) | 1.63 | 1.63 | 2.00 | 1.21 | 1.63 | 1.50 | 1.63 | 3.00 | 1.00 | 1.63 | 1.63 | 1.50 | 1.63 | 1.63 |
| Drag ocean–ice (10-3) | 10.00 | 5.00 | 1.00 | 3.24 | 5.36 | 10.00 | 5.36 | 5.50 | 5.50 | 10.00 | 5.00 | 10.00 | 5.00 | 5.00 |
| DA sea ice system | Linear nudging | None (SST) | Adjoint | None (SST) | 3DVAR | 2D local analysis SEEK filter | EnOI | None (SST) | None (SST) | 3DVAR-FGAT | OI | None (SST) | Linear nudging | Flow-dependent nudging |
| DA sea ice data | NSIDC | – | NSIDC | – | OSI-SAF | CERSAT | NSIDC | – | – | OSTIA | OSI-SAF | – | NCEP-OIv2 | NCEP-Oiv2 |
| Analysis window | 7 days | 10 days | 20 years | 1 day | 1 day | 7 days | 5 days | 1 month | 1/3 month | 5 days | 5 days | 7 days | 1 day | 1 day |

P* and Cf are parameters for the ice strength formulations following respectively Hibler (1979) and Rothrock (1975)

DA data assimilation, VP viscous-plastic, EVP elastic–viscous-plastic, SST sea surface temperature

M. Chevallier et al.

*The results discussed in the manuscript can be useful as a support of further studies using the reanalysed fields but, as it stands, the manuscript is merely descriptive. Also, little information is given about the ensemble and this information is not used to assess the quality of the reanalysis: only the ensemble mean are used for this purpose.*

**Reply**: We agree and as we answered to the other reviewer, the main objective is to present and validate the official product of Copernicus CMEMS for the Arctic region, which is provided as a deterministic reanalysis product based on the ensemble mean, for consistency with other CMEMS reanalyses. Unfortunately, our storage facility is insufficient to store the full ensemble of the daily averaged fields, and we only have at our disposal the ensemble statistics of the variables assimilated at each assimilation time (every week). We have extended our validation work with a reliability analysis (e.g. Candille et al. 2007) of the observation network assimilated according to the all assimilated variables (SST, SSH, Ice concentration, T-S, and sea ice drift).

*The quality of the reanalysis obtained using TOPAZ4 could also be compared with the quality of similar other products.*

**Reply**: We think that such comparison is beyond the scope of our paper and, for the sake of diplomatic correctness, is better undertaken in a separate collaborative initiative (The ongoing Ocean Synthesis COST action, a follow-up of the ORA-IP Arctic paper by Chevallier et al.).

A primary comparison of the ocean part of our analysis has been compared with other existing systems (Lien et al. 2016, cited in the manuscript).

**Responses to reviewers' comments**

We would like to thank once more the reviewers for their comments. The major changes in the revision are listed as following:

- Add Section 3 to assess the ensemble reliability in the TOPAZ4 reanalysis.
- Shortened the model and assimilation description to reduce overlap with Sakov et al. (2012).
- Improving the uncertainty analysis of the reanalysis with respect to in situ profiles in Section 4.
- Add figures 3, 4 and 8 into the revision
- Changes of the text in agreement with the recommendations of the reviewers.

The detailed responses are listed one by one with blue as following:

**Referee #1**

*1.* *The main purpose of the paper is the assessment of the reanalysis using all available observations. However, to compare the reanalysis to observations, the authors just compute the average and RMS difference between the ensemble mean and observations. This method looks very crude to me, and does not make justice to the advanced method that is used to perform data assimilation. The ensemble data assimilation system provides a probability distribution for the reanalysis, which is described by an ensemble of model states. Why then assessing the reanalysis using the ensemble mean only? Probabilistic tools exist to perform an objective comparison between ensemble simulations and observations (see for instance Toth et al., 2003, or Candille et al., 2007). Why performing an ensemble reanalysis if the probabilistic information is discarded to study the performance of the system? Would it be possible to include some kind of probabilistic assessment, or at least explain better why using such a crude assessment method? Would it be possible to include some kind of probabilistic assessment, or at least explain better why using such a crude assessment method?*

**Reply:** We would like to thank the reviewer for this constructive comment and suggestion. Our main purpose is to present and validate the official product of Copernicus CMEMS for the Arctic region, which is provided as a deterministic reanalysis product based on the ensemble mean, for consistency with other CMEMS reanalyses. However, we fully agree that validation of the quality of the ensemble is crucial to prove the ability of our reanalysis to make the best use of the heterogeneous observational network (spatially, temporally and various data sources); for example that we do not overfit one observational data set at the expense of the others. The reliability for an EnKF-based data assimilation system like ours is even rather important, since the efficiency of the system relies on adequate assumptions for model and observation errors. Unfortunately, our storage facility is insufficient to store the

JipingMac xie 22/12/2016 18:10
**Style Definition:** Normal (Web)

JipingMac xie 22/12/2016 18:10

JipingMac xie 22/12/2016 18:10

JipingMac xie 22/12/2016 18:10

JipingMac xie 22/12/2016 18:10

JipingMac xie 22/12/2016 18:10

JipingMac xie 22/12/2016 18:10

JipingMac xie 22/12/2016 18:10

JipingMac xie 22/12/2016 18:10
**Reply**: Yes, thank you for your comment. The comment. The modified RCRV (Talagrand et al., 1999; Candille et al., 2007) and the method of Desroziers et al. (2005) will be used to assess the concerned statistic features. .

JipingMac xie 22/12/2016 18:10

JipingMac xie 22/12/2016 18:10
**Reply**: Yes, thank you for your comment. The comment. The modified RCRV (Talagrand et al., 1999; Candille et al., 2007) and the method of Desroziers et al. (2005) will be used to assess the concerned statistic features. .

JipingMac xie 22/12/2016 18:10

JipingMac xie 22/12/2016 18:10
**Deleted: Reply**: Yes, thank you for your comment. The comment. The modified RCRV (Talagrand et al., 1999; Candille et al., 2007) and the method of Desroziers et al. (2005) will be used to assess the concerned statistic features. .

full ensemble of the daily averaged fields, and we only have at our disposal the ensemble statistics of the variables assimilated at each assimilation time (every week).

In order to address the reviewer comment, we have extended our validation work with a reliability analysis (e.g. Candille et al. 2007) of the observation network assimilated according to the all assimilated variables (SST, SSH, Ice concentration, T-S, and sea ice drift).

*2. In assessing the performance by computing the difference with observations, the paper implicitly (and sometimes explicitly) assumes that the closer to the observations, the better the reanalysis. This amount to completely neglecting observation errors in the assessment of the reanalysis, which is usually not an appropriate approximation. This incorrect assumption is for instance made explicitly in:*
- *p. 13, l. 4, where the misfit to observations is called "error" on the reanalysis;*

**Reply:** Thanks. It is corrected with "of the misfit".

- *p. 14, l. 21, where the reanalysis is said to be improved if difference to observations is smaller;*

**Reply:** Thanks. We have noted that the increased accuracy is an improvement because the reliability remained equal in the meantime.

- *p. 16, l. 6-7, where it is said that an RMSD with observations of 5% is good; whereas the accuracy of the observations is said to be about 10%. In my view, this just mean that the reanalysis is excessively close to observations.*

**Reply:** Thank you, it is corrected. The reliability analysis in Section 3 revealed - in the contrary - an underdispersion. The sentence now concentrates on the qualitative message (errors concentrated near the ice edge).

*I think that it would be important to better explain the limitations of this simple approach for assessing the performance of the reanalysis; to explain why more sophisticated comparison metrics were not applied (see my previous comment) and avoid the misleading expressions listed above.*

**Reply**: We agree with the reviewer and the above statements will be revised according to the reliability analysis. We added the reliability analyses of the modified RCRV to ensure that we are not over-fitting observations and that the ensemble does not collapse, and also use the innovation budget (Rodwell et al., 2016) to investigate the uncertainty variability in time.

JipingMac xie 22/12/2016 18:10

JipingMac xie 22/12/2016 18:10

JipingMac xie 22/12/2016 18:10

JipingMac xie 22/12/2016 18:10

JipingMac xie 22/12/2016 18:10

JipingMac xie 22/12/2016 18:10

JipingMac xie 22/12/2016 18:10

JipingMac xie 22/12/2016 18:10

JipingMac xie 22/12/2016 18:10

JipingMac xie 22/12/2016 18:10
**Reply**:

JipingMac xie 22/12/2016 18:10

JipingMac xie 22/12/2016 18:10

JipingMac xie 22/12/2016 18:10

JipingMac xie 22/12/2016 18:10

*3. In the introduction, the authors provide several arguments to support the idea that ensemble methods are an appropriate way to apply the dynamical model constraint in the estimation process. However, this is not discussed anymore in the assessment of the performance of the reanalysis. Only quantitative difference to observations are provided and analysed. I think that the quality of the paper would be enhanced if more explicit evidence of what is stated in the introduction was provided in addition to the simple description of the distance between reanalysis and observations.*

**Reply**:  This is now extensively discussed in the manuscript, both during the reliability section and in the conclusion. Despite some discontinuities causes by the change of observational data set and change in the data assimilation setting, the statistic remains relatively stable through the course of the reanalysis. The reliability budget analysis exemplifies the challenge of providing a balances reanalysis with relative contributions from various data sources.

**Referee #2**

*The manuscript appears more like a report than a scientific paper tackling a scientific or methodological issue. The model system is described elsewhere and has undergone very little changes with respect to previously published information. The assessment of the quality of the products uses a rather elementary approach.*

**Reply**: The paper by Sakov et al. (2012) was a proof of concept that an EnKF-based assimilation system can be used with a coupled ocean and sea ice for long reanalysis. This study does not propose new methodological development but it verifies that the proof of concept holds when applied for a longer period (23 years are more relevant to the community than 6 years) with a more heterogeneous observation network (spatially, temporally and various data sources). The main purpose of the manuscript is to present and validate the official Copernicus CMEMS product for the Arctic region. The proposed reanalysis is unique (see table below extracted from Chevalier et al. 2016) as it proposes a long high-resolution dynamical reconstruction of the ocean and sea ice, and assimilates a complete set of observations available in the Arctic region with an advanced ensemble data assimilation method and with strongly coupled data assimilation between ocean and sea-ice. We have tried to present this achievement in a concise manner, with a primary focus to inform the end-user about the strength and weaknesses of our data set. As a response to the recommendation of the first reviewer (and your following comment), we have extended the current

JipingMac xie 22/12/2016 18:10

JipingMac xie 22/12/2016 18:10

JipingMac xie 22/12/2016 18:10

JipingMac xie 22/12/2016 18:10

JipingMac xie 22/12/2016 18:10

JipingMac xie 22/12/2016 18:10
**Deleted: Reply**: Thanks for this comment. However, we disagree this point and the reasons are list as follow: 1) The TOPAZ4 reanalysis is an important new reanalysis product describing the Arctic Ocean (delivered by the Arctic component of the MyOcean system). Although there are some documents (like Sakov et al., 2012) showing the part of validation results, the complete and detailed assessment is still unknown for the public. 2) The reanalysis products in the Arctic with long time period are very few, and their qualities about the dataset self have not been revealed clearly for both ocean and sea ice states. The assessment of the TOPAZ4 reanalysis can partly answer how about the present reanalysis quality in the Arctic. 3) Based on the present assessment results in this paper, we can find some weaknesses which are useful for the concerned system developer. 4) The model system has been simply described and is basic background for understanding the reanalysis system, although it had been described elsewhere.

validation with the analysis of the ensemble reliability, and asses whether our system manage to provide a dynamical reconstruction that falls within the uncertainty of the different observational data sets that are assimilated. We believe it has increased the scientific value of our manuscript.

**Table 1** System configuration and selected parameters

| Name | C-GLORS05 | CNRM | ECCO-v4 | ECDA | GloSea5 | G2V3 | MERRA Ocean | MOVE-CORE | MOVE-G2 | ORAP5 | UR025.4 | G2V1 | ERA L | ER AN |
|---|---|---|---|---|---|---|---|---|---|---|---|---|---|---|
| Institution | CMCC | CNRM-GAME | JPL/NASA, MIT, AER | GFDL/NOAA | UK Met Office | Mercator Ocean | GSFC/NASA/GMAO | MRI/JMA | MRI/JMA | ECMWF | University of Reading | Mercator Océan | ECMWF | ECMWF |
| Nominal horizontal resolution | 0.5° | 1° | 0.4°–1.0° | 1° | 0.25° | 0.25° | 0.5° | 0.5° × 1° | 0.3–0.5° × 1° | 0.25° | 0.25° | 0.25° | 1° | 1° |
| Ocean–sea ice model | NEMO3.2-LIM2 | NEMO3.2-GELATO5 | MITgcm | GFDL-MOM4.4.1-SIS | NEMO3.2-CICE4.0 | NEMO3.1-LIM2 (EVP) | MOM4.1-CICE4.0 | MRI COM3-Mellor & Kanta + CICE4.0 | MRI.COM3-Mellor & Kanta + CICE4.0 | NEMO3.4-LIM2 | NEMO3.2-LIM2 | NEMO3.1-LIM2 (EVP) | NEMO3.2-LIM2 | NEMO3.2-LIM2 |
| Time period | 1979–2011 | 1990–2010 | 1992–2010 | 1961–2014 | 1993–2011 | 1993–2011 | 1979–present | 1948–2007 | 1993–2012 | 1979–2012 | 1989–2010 | 1993–2009 | 1990–2011 | 1990–2011 |
| Source of atmospheric forcing data | ERA-Interim | ERA-Interim | ERA-Interim | Coupled run constrained to NCEP/NCAR-NCEP/DOE | ERA-Interim | ERA-Interim | Coupled run constrained to MERRA | CORE | JRA55 | ERA-Interim | ERA-Interim | ERA-Interim | ERA-Interim | ERA-Interim |
| Vertical discretization | 2 ice + 1 snow | 9 ice + 1 snow | 1 ice + 1 snow | 2 ice + 1 snow | 1 ice + 1 snow | 2 ice + 1 snow | 4 ice + 1 snow | 1 ice + 1 snow | 1 ice + 1 snow | 2 ice + 1 snow | 2 ice + 1 snow | 2 ice + 1 snow | 2 ice + 1 snow | 2 ice + 1 snow |
| Thickness categories | 1 | 8 | 1 | 5 | 5 | 1 | 5 | 5 | 1 | 1 | 1 | 1 | 1 | 1 |
| Dynamics | EVP | EVP | VP | EVP | EVP | EVP | EVP | EVP | EVP | VP | VP | EVP | VP | VP |
| $P^*$ (N/m)/Cf (–) | $P^* = 2.0 \times 10^4$ | $P^* = 2.75 \times 10^4$ | $P^* = 2.754 \times 10^4$ | $P^* = 2.5 \times 10^4$ | $Cf = 17$ | $P^* = 2 \times 10^4$ | $P^* = 2.75 \times 10^4$ | $P^* = 2.75 \times 10^4$ | $P^* = 2.75 \times 10^4$ | $P^* = 1.50 \times 10^4$ | $P^* = 1 \times 10^4$ | $P^* = 2 \times 10^4$ | $P^* = 1.5 \times 10^4$ | $P^* = 1.5 \times 10^4$ |
| Drag air–ice (10-3) | 1.63 | 1.63 | 2.00 | 1.21 | 1.63 | 1.50 | 1.63 | 3.00 | 1.00 | 1.63 | 1.63 | 1.50 | 1.63 | 1.63 |
| Drag ocean–ice (10-3) | 10.00 | 5.00 | 1.00 | 3.24 | 5.36 | 10.00 | 5.36 | 5.50 | 5.50 | 10.00 | 5.00 | 10.00 | 5.00 | 5.00 |
| DA sea ice system | Linear nudging | None (SST) | Adjoint | None (SST) | 3DVAR | 2D local analysis SEEK filter | EnOI | None (SST) | None (SST) | 3DVAR-FGAT | OI | None (SST) | Linear nudging | Flow-dependent nudging |
| DA sea ice data | NSIDC | – | NSIDC | – | OSI-SAF | CERSAT | NSIDC | – | – | OSTIA | OSI-SAF | – | NCEP-OIv2 | NCEP-OIv2 |
| Analysis window | 7 days | 10 days | 20 years | 1 day | 1 day | 7 days | 5 days | 1 month | 1/3 month | 5 days | 5 days | 7 days | 1 day | 1 day |

$P^*$ and Cf are parameters for the ice strength formulations following respectively Hibler (1979) and Rothrock (1975)

*DA* data assimilation, *VP* viscous-plastic, *EVP* elastic–viscous-plastic, *SST* sea surface temperature

M. Chevallier et al.
* * *
*The results discussed in the manuscript can be useful as a support of further studies using the reanalysed fields but, as it stands, the manuscript is merely descriptive. Also, little information is given about the ensemble and this information is not used to assess the quality of the reanalysis: only the ensemble mean are used for this purpose.*

**Reply**: We agree and as we answered to the other reviewer, the main objective is to present and validate the official product of Copernicus CMEMS for the Arctic region, which is provided as a deterministic reanalysis product based on the ensemble mean, for consistency with other CMEMS reanalyses. Unfortunately, our storage facility is insufficient to store the full ensemble of the daily averaged fields, and we only have at our disposal the ensemble statistics of the variables assimilated at each assimilation time (every week). We have extended our validation work with a reliability analysis (e.g. Candille et al. 2007) of the observation network assimilated according to the all assimilated variables (SST, SSH, Ice concentration, T-S, and sea ice drift).

*The quality of the reanalysis obtained using TOPAZ4 could also be compared with the quality of similar other products.*

**Reply**: We think that such comparison is beyond the scope of our paper and, for the sake of diplomatic correctness, is better undertaken in a separate collaborative initiative (The ongoing Ocean Synthesis COST action, a follow-up of the ORA-IP Arctic paper by Chevallier et al.).

A primary comparison of the ocean part of our analysis has been compared with other existing systems (Lien et al. 2016, cited in the manuscript).

[revised manuscript text omitted]

xie 22/12/2016 17:59

xie 22/12/2016 17:59

xie 22/12/2016 17:59

xie 22/12/2016 17:59

xie 22/12/2016 17:59

xie 22/12/2016 17:59

xie 22/12/2016 17:59

[revised manuscript text omitted]

xie 22/12/2016 17:59

xie 22/12/2016 17:59

xie 22/12/2016 17:59

xie 22/12/2016 17:59